# Developmental single-cell transcriptomics of hypothalamic POMC neurons reveal the genetic trajectories of multiple neuropeptidergic phenotypes

**Hui Yu[1]\*, Marcelo Rubinstein[1,2,3]\*, Malcolm J Low[1]\***

[1]Department of Molecular and Integrative Physiology, University of Michigan Medical School, Ann Arbor, United States; [2]Instituto de Investigaciones en Ingeniería Genética y Biología Molecular, Consejo Nacional de Investigaciones Científicas y Técnicas, Buenos Aires, Argentina; [3]Departamento de Fisiología, Biología Molecular y Celular, Facultad de Ciencias Exactas y Naturales, Universidad de Buenos Aires, Buenos Aires, Argentina

**\*For correspondence:**
huiyuz@umich.edu (HY);
mrubins@dna.uba.ar (MR);
mjlow@umich.edu (MJL)

**Competing interest:** The authors declare that no competing interests exist.

**Abstract** Proopiomelanocortin (POMC) neurons of the hypothalamic arcuate nucleus are essential to regulate food intake and energy balance. However, the ontogenetic transcriptional programs that specify the identity and functioning of these neurons are poorly understood. Here, we use single-cell RNA-sequencing (scRNA-seq) to define the transcriptomes characterizing *Pomc*-expressing cells in the developing hypothalamus and translating ribosome affinity purification with RNA-sequencing (TRAP-seq) to analyze the subsequent translatomes of mature POMC neurons. Our data showed that *Pomc*-expressing neurons give rise to multiple developmental pathways expressing different levels of *Pomc* and unique combinations of transcription factors. The predominant cluster, featured by high levels of *Pomc* and *Prdm12* transcripts, represents the canonical arcuate POMC neurons. Additional cell clusters expressing medium or low levels of *Pomc* mature into different neuronal phenotypes featured by distinct sets of transcription factors, neuropeptides, processing enzymes, cell surface, and nuclear receptors. We conclude that the genetic programs specifying the identity and differentiation of arcuate POMC neurons are diverse and generate a heterogeneous repertoire of neuronal phenotypes early in development that continue to mature postnatally.

## Editor's evaluation

This study investigates the developmental origins of functionally distinct neuronal populations in the arcuate nucleus of the hypothalamus. This study reports the transcriptional program of develop of multiple subtypes of POMC neurons, which are important for multiple physiological and behavioral functions, including appetite. The paper uses cutting edge methods will be important for research studying the arcuate nucleus including regulating food intake and metabolism.

## Introduction

*Proopiomelanocortin (POMC)*-expressing neurons located in the arcuate nucleus of the hypothalamus play an essential role in the regulation of food intake by maintaining a melanocortin-dependent anorexigenic tone. The critical importance of hypothalamic melanocortins is apparent in mice lacking *Pomc* that develop hyperphagia and early-onset extreme obesity (*Bumaschny et al., 2012*). This

phenotype is closely mirrored in humans carrying biallelic null mutations in *POMC* (***Krude et al., 1998***). In addition, arcuate POMC neurons release the *Pomc*-encoded opioid peptide β-endorphin, which is critical in mediating normal food intake (***Appleyard et al., 2003***) and stress-induced analgesia (***Rubinstein et al., 1996***).

Postmitotic POMC neurons acquire their phenotypic identity in the developing hypothalamus days earlier than any other peptidergic neurons born in the presumptive arcuate nucleus (e.g. NPY/AGRP, GHRH, SST, KISSPEPTIN). *Pomc* mRNA is initially observed in the tuberal portion of the prospective mouse hypothalamus of E10.5 embryos, following the concurrent advent of the essential transcription factors (TFs) ISL1 (***Nasif et al., 2015***), NKX2-1 (***Orquera et al., 2019***), and PRDM12 (***Hael et al., 2020***) that act together in a combinatorial manner. In fact, the early ablation of either *Isl1*, *Nkx2-1*, or *Prdm12* disrupts the onset of hypothalamic *Pomc* expression in conditional mutant mouse embryos (***Nasif et al., 2015***; ***Orquera et al., 2019***; ***Hael et al., 2020***). Given that the number of arcuate neurons co-expressing *Nkx2-1*, *Isl1*, and *Prdm12* greatly exceeds that of POMC neurons in this area, it is expected that additional and still unknown TFs integrate a core regulatory complex completing the genetic program that specifies the identity of POMC neurons.

Other molecular markers of arcuate POMC neurons such as transporters, receptors, channels, and co-transmitters exhibit great variability indicating that a heterogeneous pool of diverse subpopulations of these neurons are involved in multiple circuits and functions. In fact, previous studies have suggested that subsets of arcuate *Pomc*-expressing neurons may act as precursors of terminally differentiated neurons of alternative neuropeptidergic phenotypes such as NPY/AGRP and KISSPEPTIN/NEUROKININ-B/DYNORPHIN (KNDY) neurons in the adult hypothalamus (***Padilla et al., 2010***; ***Sanz et al., 2015***). Initial attempts to characterize the transcriptome of adult POMC neurons at the single-cell level confirmed the heterogeneous nature of arcuate POMC neurons and revealed the existence of distinct clusters including those co-expressing the aforementioned neuropeptide genes (***Campbell et al., 2017***; ***Chen et al., 2017***; ***Lam et al., 2017***; ***Huisman et al., 2020***). However, the molecular signatures of the various hypothalamic *Pomc*-expressing lineages are still lacking.

Here, we track the origin and maturation of arcuate POMC neurons in the developing and early postnatal hypothalamus by performing single-cell RNA-sequencing (scRNA-seq) transcriptomics of fluorescently labeled cells taken from *Pomc-TdDiscomaRed-Sv40PolyA* (*Pomc-TdDsRed*) transgenic mice at different embryonic (E11.5, E13.5, E15.5, and E17.5) and early postnatal (P5 and P12) ages. Our data indicate that POMC neurons give rise to distinct transcriptional trajectories and show that a heterogeneous population of *Pomc*-expressing neurons is established early in the developing arcuate nucleus. Combining translating ribosome affinity purification with RNA-sequencing (TRAP-seq) of *Pomc-eGFPL10a* transgenic mice at P12 and P60, we identified transcriptional programs that emerge from distinct populations of early embryonic POMC neurons that later mature into fully functional POMC neurons. We believe that these datasets provide a valuable resource for comprehensively understanding the genetic bases underlying the ontogeny and development of POMC neurons as well as for the rational design of further functional studies of brain circuits involving POMC neurons.

## Results

### Cluster analysis of gene expression in POMC positive hypothalamic cells integrated across six development ages

We captured the single-cell transcriptomes of hypothalamic *Pomc*-expressing cells at four embryonic days (E11.5, E13.5, E15.5, and E17.5) and two early postnatal days (P5 and P12) when the maturing neurons already develop axons and innervate key target nuclei distal to the Arc (***Bouret, 2004a***, ***Bouret et al., 2004b***, ***Bouret et al., 2012***). Medial basal hypothalami from mice that express the reporter transgene *Pomc-TdDsRed* selectively in POMC cells were pooled from multiple embryos or postnatal pups at the indicated ages. Dissociated single-cell suspensions were sorted for DsRed fluorescence and used for library preparation with the 10× Genomics platform (***Figure 1A***). After a rigorous screening procedure to remove substandard scRNA-seq data (see Materials and methods), we used a combination of computational platforms to analyze the transcriptomes of 13,962 cells that had *Pomc* counts of ≥1 unique molecular identifiers (UMIs).

A Seurat analysis of all cells integrated across the six time points revealed 11 distinct cell clusters projected onto a UMAP plot (***Figure 1B***). The same data are represented in a heatmap corresponding

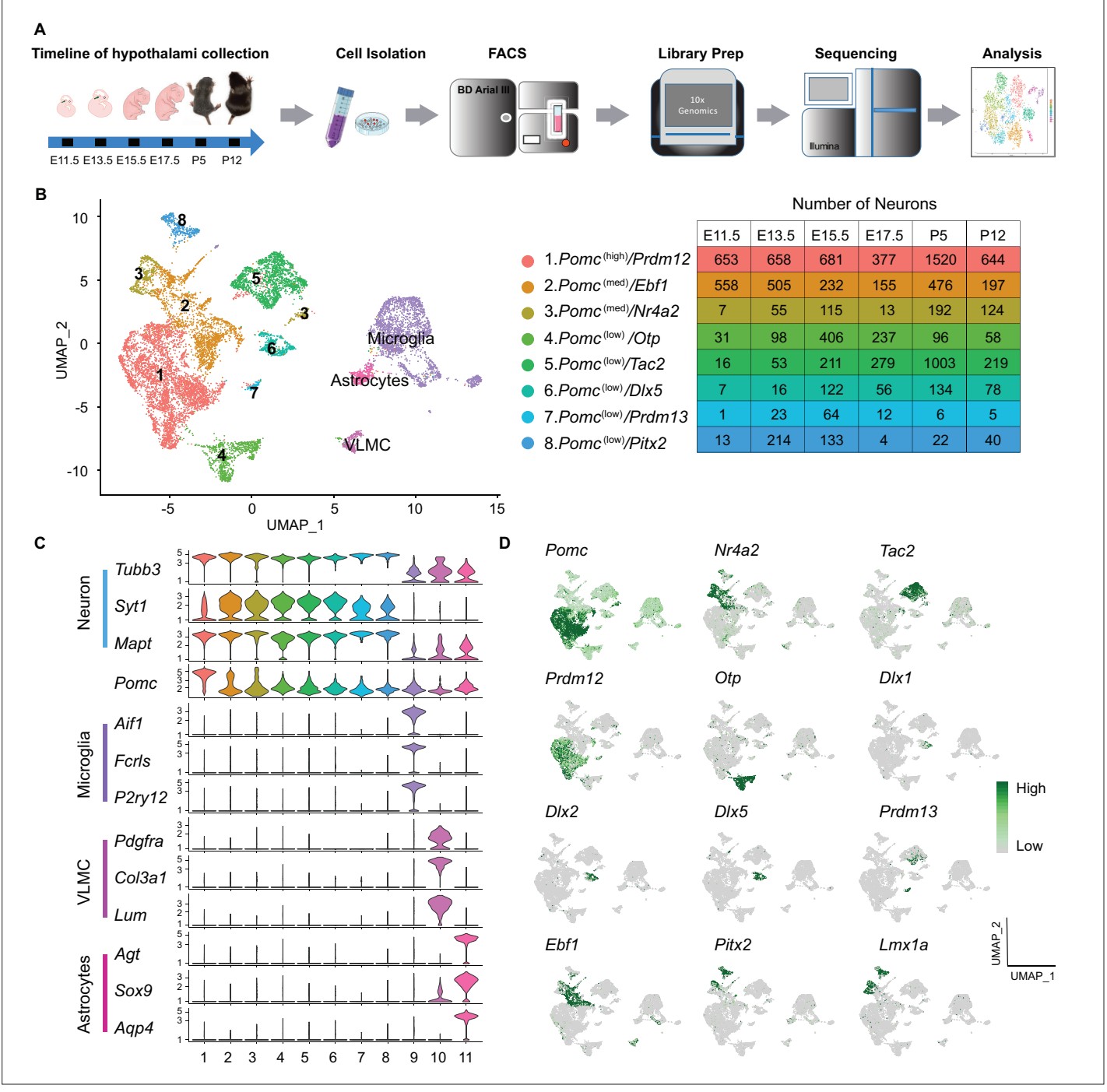

**Figure 1.** Single-cell sequencing of *Pomc*^DsRed^ cells from developing mouse hypothalami at E11.5, E13.5, E15.5, E17.5, P5, and P12 reveals eight distinct neuronal clusters and three non-neuronal clusters. (**A**) Schematic diagram showing the overall experimental procedure. (**B**) UMAP (Uniform Manifold Approximation and Projection) plot showing the distribution of major cell clusters and their corresponding numbers of cells at each developmental stage. (**C**) Violin plots showing the expression of three signature genes and *Pomc* for each cluster. (**D**) UMAP plots showing the enrichment of feature genes in each cluster. Genes representing each cluster were colored and highlighted to show the cluster-specific enrichment. The color intensity corresponds to the normalized gene expression from Seurat analysis, where gene expression for each cell was normalized by the total expression, multiplied by a scale factor of 10,000, and then log-transformed. VLMC, vascular leptomeningeal cell.

The online version of this article includes the following source data and figure supplement(s) for figure 1:

**Source data 1.** Average gene expression obtained from the Seurat analysis of each of the 11 proopiomelanocortin (*Pomc*) clusters integrated across all six developmental stages (corresponding to *Figure 1*).

*Figure 1 continued on next page*

*Figure 1 continued*

**Source data 2.** Feature genes that define each of the 11 proopiomelanocortin (*Pomc*) clusters (corresponding to *Figure 1*) integrated across all six developmental stages.

**Figure supplement 1.** Heatmap showing the expression of the top 30 marker genes defining each cluster from *Figure 1*.

**Figure supplement 2.** Analysis of proopiomelanocortin (POMC) cells with *DsRed* transcripts reveals nine distinct cell clusters.

**Figure supplement 2—source data 1.** Average gene expression obtained from the Seurat analysis of each of the nine DsRed clusters integrated across all six developmental stages (corresponding to *Figure 1—figure supplement 2*).

**Figure supplement 3.** Overview of cell clusters from proopiomelanocortin (POMC) cells with *DsRed* transcripts.

**Figure supplement 3—source data 1.** Feature genes that define each of the nine DsRed clusters integrated across all six developmental stages (corresponding to *Figure 1—figure supplement 3*).

to the top 30 expressed genes per cluster (*Figure 1—figure supplement 1*). Clusters 1–8 (10,828 cells) were all highly enriched for a panel of three neuronal marker gene transcripts. The embedded table in *Figure 1B* lists the number of neurons of each cluster at each developmental age. The other three clusters (3134 cells) were enriched in transcripts characteristic of either microglia, vascular leptomeningeal cells (VLMC), or astrocytes (*Figure 1C*). Gene ontology analyses for biological processes further confirmed the neuronal or non-neuronal identities of the 11 individual clusters (data not shown).

By definition, all cells in the neuronal and non-neuronal clusters contained *Pomc* transcripts (UMI ≥1), however there were significant differences in the average expression levels per cluster, ranging from 1.3 to 33.8 [exp(*log Normalized* counts) –1]. Nomenclature of the eight neuronal clusters is based on a combination of relatively high (>20), medium (5–20), or low (<5) *Pomc* levels and high expression of one differential key feature gene as follows: no. 1, $Pomc^{(high)}/Prdm12$; no. 2 $Pomc^{(med)}/Ebf1$; no. 3, $Pomc^{(med)}/Nr4a2$; no. 4, $Pomc^{(low)}/Otp$; no. 5, $Pomc^{(low)}/Tac2$; no. 6, $Pomc^{(low)}/Dlx5$; no. 7, $Pomc^{(low)}/Prdm13$; and no. 8, $Pomc^{(low)}/Pitx2$. The cluster distribution and relative expression of those feature genes are represented by individual UMAP plots (*Figure 1D*). Average expression levels of all genes, including *Pomc*, in the 11 clusters and complete lists of all feature genes that define the individual clusters are presented in *Figure 1—source data 1* and *Figure 1—source data 2*, respectively.

In order to validate our experimental approach of isolating *Pomc*-expressing cells from the developing hypothalamus by fluorescence-activated cell sorting (FACS) for the transgenic surrogate marker *Pomc-TdDsRed*, we performed an independent unsupervised cluster analysis of the transcriptomes from the 5724 cells that had *DsRed* counts of ≥1 UMI (*Figure 1—figure supplement 2A-C* and ). A projection analysis (*Figure 1—figure supplement 2D-F*) of the differentially expressed feature genes defined nine DsRed clusterswhich closely corresponded to 9 of the 11 clusters based on *Pomc* counts of ≥1 UMI. The two exceptions were the smallest clusters 7. $Pomc^{(low)}/Prdm13$ and astrocytes (*Figure 1—figure supplement 3D-F*). Furthermore, the number of cells at each developmental age from the corresponding neuronal clusters was remarkably similar in the *Pomc* UMI ≥ 1 dataset compared to the *DsRed* UMI ≥ 1 dataset (compare *Figure 1B* to *Figure 1—figure supplement 3A*). The different population of DsRed fluorescent sorted cells showing UMI ≥ 1 for *Pomc* (13,962) or *DsRed* (5724) may be due to differential half-lives of the fluorescent protein DSRED used during cell sorting and its transcript used for barcoding the RNA-seq libraries.

## Temporal gene expression analysis of the eight neuronal clusters at individual developmental stages

Next, we analyzed temporal gene expression patterns for each of the eight neuronal clusters at the six distinct developmental ages (*Figure 2—source data 1*). The progressive changes in feature genes across time are responsible for the observed within cluster heterogeneity on the heatmap in *Figure 1—figure supplement 1*.

### Cluster 1

$Pomc^{(high)}/Prdm12$ cells were the most abundant among the eight neuronal clusters and consistently had the highest levels of *Pomc* expression at all six developmental ages. We consider them to be the cell lineage that matures into the canonical POMC neurons in the adult hypothalamus (*Figure 2A–C*). Other feature genes included a set of four TFs known to be critical for the initiation and/or maintenance of Arc *Pomc* expression, *Isl1*, *Nkx2-1*, *Prdm12*, and *Tbx3* (*Nasif et al., 2015*; *Orquera et al.,*

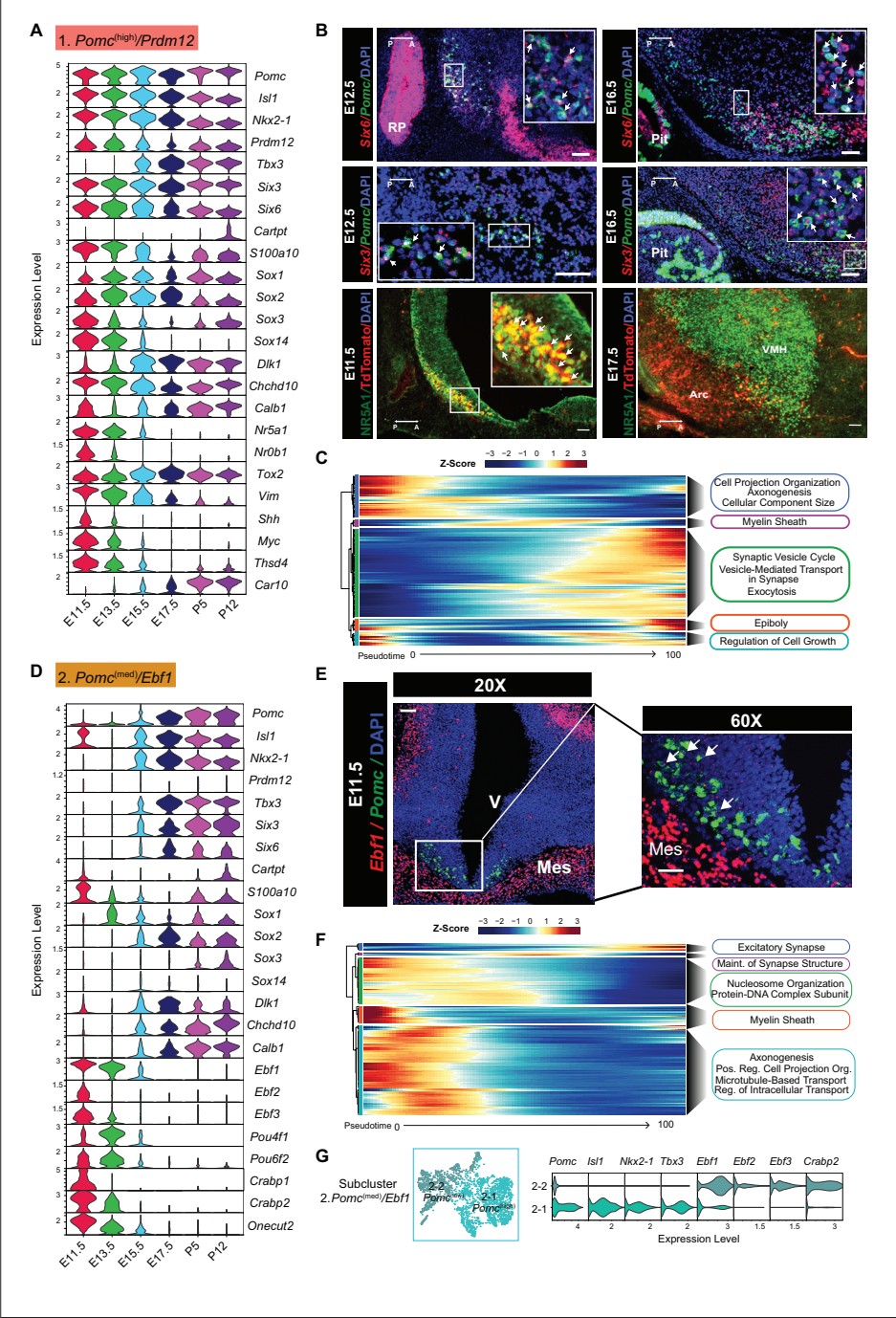

**Figure 2.** Temporal gene expression patterns of the *Pomc*(high)/*Prdm12* cluster and *Pomc*(med)/*Ebf1* clusters. (**A** and **D**) Violin plots showing the expression of signature genes at each developmental stage in the *Pomc*(high)/*Prdm12* cluster and *Pomc*(med)/*Ebf1* cluster, respectively. (**B**) Fluorescence in situ hybridization showing the co-localization of *Pomc* (green) and *Six6* (red), the co-localization of *Pomc* (green) and *Six3* (red) at indicated developmental stages. Immunofluorescence showing the co-localization TDTOMATO (red) NR5A1 (green) at the indicated developmental stages. (**C** and **F**) Heatmaps showing the gene ontology analysis based on the top marker genes in the order of pseudotime in *Pomc*(high)/*Prdm12* cluster and *Pomc*(med)/*Ebf1* cluster, respectively. (**E**) Fluorescence in situ hybridization showing the co-localization of *Pomc* (green) and *Ebf1* (red) at E11.5. (**G**) Reclustering cells from the *Pomc*(med)/*Ebf1* cluster reveals two subclusters denoted as 2–1 *Pomc*(high) and 2–2 *Pomc*(low). Violin plots showing the signature gene expression in 2–1 *Pomc*(high) and 2–2 *Pomc*(low) clusters. Insets are magnified views of the indicated boxes. Arrows indicate co-expressing neurons in the merged panels. Scale bar: 50 μm. Image orientation: left,

*Figure 2 continued on next page*

*Figure 2 continued*

posterior(P); right, anterior(A). RP: Rathke's pouch; Pit: pituitary gland. Arc: arcuate nucleus; VMH: ventromedial hypothalamus; V: ventricular zone; Mes: mesenchyme.

The online version of this article includes the following source data and figure supplement(s) for figure 2:

**Source data 1.** Average gene expression obtained from the Seurat analysis of six developmental stages of each cluster (corresponding to *Figure 2*, *Figure 3*, and their linked *Figure 2—figure supplement 2*).

**Source data 2.** Average gene expression obtained from the Seurat analysis of the two subclusters 3–1 and 3–2 derived from reclustering of cluster 3.

**Source data 3.** Feature genes that define each of the two subclusters 3–1 and 3–2 derived from reclustering of cluster 3.

**Figure supplement 1.** Fluorescence in situ hybridization showing the co-localization of *Pomc* (green) and *Ebf1* (red) at E13.5 (**A**) and at E17.5 (**B**) and the co-localization of *Pomc* (green) and *Pou4f1* (red) at E13.5 (**C**) and at E17.5 (**D**), respectively.

**Figure supplement 2.** Temporal gene expression patterns of the *Pomc*<sup>(med)</sup>/*Nr4a2* and *Pomc*<sup>(low)</sup>/*Pitx2* clusters.

---

*2019*; *Quarta et al., 2019*, *Hael et al., 2020*; *Figure 2A*). They were expressed in similar developmental patterns to each other, with the exception of *Tbx3* that was significantly expressed only from day E15.5. Notably, there was a lack of the pituitary-specific *Tbx19* TF, confirming that no pituitary tissue was included in the hypothalamic dissections. Two homologues of the *Drosophila sine oculis* homeobox gene, *Six3* and *Six6,* which have not been described previously in the context of Arc POMC neuron development, were also expressed robustly at all six time points (*Figure 2A*). Co-expression of their mRNAs with *Pomc* by RNAScope at ages E12.5 and E16.5 confirmed the scRNA-seq analysis, although the two TFs were expressed in additional hypothalamic domains and the pituitary gland (*Figure 2B*). *Vim1*, *Sox1*, *-2, -3,* and *-14,* and *Myc*, whose expression is typically associated with neuronal precursors or immature neurons, tended to have higher expression at the earlier embryonic ages (*Figure 2A*). The nuclear receptor gene *Nr5a1*, also known as steroidogenic factor 1 (*Sf1*), was a key feature gene expressed almost exclusively at age E11.5 (*Figure 2A*), but only minimally expressed at later embryonic time points when NR5A1 is critically involved in the development of the hypothalamic ventromedial nucleus (*Ikeda et al., 1995*). This differential spatio-temporal presence of NR5A1 within or outside POMC neurons was confirmed by immunofluorescence on hypothalamic sections taken from E11.5 and E17.5 mouse embryos (*Figure 2B*). Interestingly, *Nr5a1*, *Nr0b1* (*Dax1*), and *Shh* transcripts were also present in cluster 1 at the two earliest ages, and are factors known to interact with the *Wnt*/β-catenin signaling pathway in early cell fate determination in the adrenal and pituitary glands (*Luo et al., 1994*; *Meeks et al., 2003*; *Mizusaki et al., 2003*). A combination of pseudotime and gene ontology analyses for the top expressed genes (Log2FoldChange > 0.25, p < 0.05) in cluster 1 revealed that the genes expressed at the beginning of pseudotime function in the organization of cell projections, axonogenesis, and the regulation of cellular component size, whereas genes expressed at the middle and end of the pseudotime function are related to neuron maturation and neuron-specified functions such as the formation of myelin sheath and regulation of synaptic vesicle cycling (*Figure 2C*).

## Cluster 2

*Pomc*<sup>(med)</sup>/*Ebf1* neurons made up the second highest abundance cluster of POMC neurons, particularly at E11.5 and E13.5. After these time points, the fraction of all neurons constituting cluster 2 dropped by approximately two-thirds from ages E15.5 through P12. In contrast, the number of neurons in clusters 3–8 increased over time with peaks at different embryonic ages (*Figure 1B*). Cluster 2 neurons, on average, had medium levels of *Pomc* transcripts that were divided distinctly from their lowest at ages E11.5–E13.5 to highest at later embryonic and postnatal ages (*Figure 2D*; *Figure 2—source data 1*). Major cluster 2 feature markers included members of the non-basic helix-loop-helix early B-cell TF family EBF (*Figure 1—source data 2*). *Ebf1* is typically co-expressed with either *Ebf2* or *Ebf3* and pairs of these factors were shared not only in cluster 2, but also in clusters 3 and 8 (*Figure 2—figure supplement 2*). Dual RNAScope in situ hybridization showed overlapping co-expression of *Ebf1* and *Pomc* mRNAs in the ventricular zone of the developing hypothalamus at an early embryonic stage (E11.5) (*Figure 2E*). However, transcript levels of *Ebf1* are much lower than what was found in

other areas of the same sections, such as the mesenchyme (MES) (*Figure 2E*). *Ebf1* mRNA was also detected in *Pomc*-expressing cells at E13.5, and then was barely detectable at E17.5 (*Figure 2—figure supplement 1A and B*). A similar expression pattern was found with another cluster 2 featured gene, *Pou4f1* (*POU Class 4 Homeobox 1*). Dual RNAScope in situ hybridization confirmed the overlap of *Pou4f1* and *Pomc* mRNAs at E13.5, but not at E17.5 (*Figure 2—figure supplement 1C and D*). These results, together, suggest that *Pomc* mRNA levels peak only after expression of both *Ebf1* and *Pou4f1* decrease. Two other prominent feature genes in cluster 2 were *Crabp1* and *Crabp2*, which encode cellular retinoic acid binding proteins. A majority of the marker genes for cluster 2 are predicted to be expressed at the beginning of pseudotime and the main functions are associated with the maintenance of basic cell biological processes such as nucleosome organization, axonogenesis, microtubule transport, and intracellular transport (*Figure 2F*). The genes ordered in the middle and end of pseudotime function for the regulation of synapse structure and activities.

Visual examination of the UMAP plots in *Figure 1D* suggested that cluster 2 was composed of two compartments, one with high and one with low *Pomc* abundance. Unsupervised reclustering of the differentially expressed genes from integrated cluster 2 confirmed the existence of two subclusters (*Figure 2G*). Subcluster 2–1 contained two-thirds of the neurons, which were POMC(high) whereas subcluster 2–2 contained the other one-third of cluster 2 neurons, which in contrast, were *Pomc*(low). As shown in *Figure 2D*, subcluster 2–2 contains cells mainly from the early developmental stages E11.5 to E15.5, whereas, subcluster 2–1 contains cells from E17.5 to P12. Consistently, known activating TFs for *Pomc* expression such as *Nkx2-1*, *Isl1*, and *Tbx3* were limited to subcluster 2–1 while relatively high levels of *Ebf* and *Crabp* transcripts were characteristic of subcluster 2–2 (*Figure 2—source data 2* and *Figure 2—source data 3*). Another distinctive feature of cells in cluster 2 (both subclusters) is the absence of *Prdm12* transcripts, one main component of cluster 1 (*Figure 2A and D*).

## Cluster 3

*Pomc*(med)/*Nr4a2* neurons displayed high expression of its major feature gene *Nr4a2* (Nuclear Receptor Subfamily 4 Group A Member 2) (*Figure 2—figure supplement 2*). *Nr4a2* mRNA was also expressed at relatively high levels in clusters 2. *Pomc*(med)/*Ebf1* and 8. *Pomc*(low)/*Pitx2. Nr4a2* (*Nurr1*) has been implicated in the regulation of *Pomc* expression in the pituitary corticotropic cell line AtT20 (*Kovalovsky et al., 2002*). Furthermore, *Nr4a2* is a pioneer transcriptional regulator important for the differentiation and maintenance of meso-diencephalic dopaminergic (mdDA) neurons during development (*Perlmann and Wallén-Mackenzie, 2004*). It is crucial for expression of a set of genes including *Slc6a3* (plasma membrane dopamine transporter, DAT), *Slc18a2* (synaptic vesicular mono-amine transporter, VMAT), *Th* (tyrosine hydroxylase), *Foxa1* and *Foxa2* (forkhead family TFs), and *Drd2* (dopamine receptor 2) that together define the dopaminergic neuronal phenotype (*Hong et al., 2014*; *Jankovic et al., 2005*; *Saucedo-Cardenas et al., 1998*; *Smits et al., 2003*). An additional key factor involved in the generation and maintenance of mdDA neurons, *Lmx1a* (Lim- and homeodomain factor 1a) (*Andersson et al., 2006*; *Hong et al., 2014*), was also a key feature gene of clusters 3 and 8 (*Figure 2—figure supplement 2A and D*) suggesting that POMC and DA neurons share at least part of their genetic differentiation programs.

Similar to cluster 2, reclustering of the differentially expressed genes from cluster 3 revealed two subclusters (*Figure 2—figure supplement 2C*). Subcluster 3–1 constituted 72% of the main cluster 3 and was characterized by *Pomc*(low) neurons with strong *Nr4a2* expression at all developmental ages. Feature genes for subcluster 3–1 also included *Lmx1a*, *Slc6a3*, *Slc18a2, Foxa2*, and *Th*, all feature genes for developmental subclusters E13.5–10 and E15.5–11 derived from cluster 3 indicating that a fraction of cluster 3 neurons were dopaminergic and existed transiently during a 3-day developmental window (*Figure 1—source data 2*, *Figure 2—source data 1*). In contrast, subcluster 3–2 neurons were *Pomc*(high) together with characteristic expression of the known *Pomc* transcriptional activator genes and were present only at ages P5 and P12. The dissociation of high *Pomc* expression with low *Nr4a2* expression in subcluster 3–2 suggests that the latter is not necessary for hypothalamic *Pomc* expression, unlike its role in pituitary corticotrophs (*Murphy and Conneely, 1997*).

## Cluster 8

*Pomc*(low)/*Pitx2* is more closely related to clusters 2. *Pomc*(med)/*Ebf1* and 3. *Pomc*(med)/*Nr4a2* than to any of the others (*Figure 2—figure supplement 2D*). In addition to *Pitx2*, major feature genes were

*Barhl1, Foxa1, Foxp2, Lmx1a/b,* and *Tac1* (*Figure 1—source data 2*). Intriguingly, virtually all of the top cluster 8 defining genes (*Figure 1—source data 2*) are identical to those that characterize both the mesencephalic dopamine neurons and glutamatergic, non-dopaminergic, subthalamic nucleus neurons (*Kee et al., 2017*). Dual RNAscope in situ hybridization for *Pitx2* and *Pomc* showed limited overlap at E12.5 (*Figure 2—figure supplement 2E*). Cluster 8 is the only cluster with the neuropeptide *Cck* as a feature gene at developmental ages P5 and P12 (*Figure 1—source data 1* and *Figure 2—source data 1*).

## Cluster 4

*Pomc*(low)*/Otp* cells uniquely expressed *Otp* at all developmental ages (*Figure 3A*). *Pomc* expression was moderate at E11.5 and then dropped to consistently lower levels through age P12 despite continued expression of *Isl1, Nkx2-1,* and *Tbx3*. Interestingly, expression of the other transcriptional modulator important for *Pomc* expression, *Prdm12*, was detected only at E11.5. Other characteristic features of cluster 4 were increasing gradients of *Agrp, Npy,* and *Calcr* expression over the course of development (*Figure 3A*). Minimal co-expression of *Pomc* and *Npy* was confirmed by fluorescence in situ hybridization (FISH) at age E15.5 (*Figure 3B*). *Sst* was highly expressed in this cluster at ages E13.5, P5, and P12 with a temporary drop off at E15.5 and E17.5. Altogether, these patterns of gene expression indicate that a subpopulation of POMC neurons expressing *Otp* further differentiates into AGRP/NPY neurons as suggested previously by studies based on transgenic lineage tracing (*Padilla et al., 2010*). Most of the top enriched genes of this cluster are expressed in the middle and end portions of pseudotime prediction highlighting functions such as regulation of inhibitory synapses, ribosomal biogenesis, and synaptic vesicle cytoskeletal transport (*Figure 3C*). A map of gene expression in individual cells within the cluster 4 UMAP plot demonstrated that *Npy* was expressed in virtually all *Agrp* neurons, but few *Sst* neurons. *Sst* and *Agrp* neurons were essentially separate cell populations (*Figure 3D*). The GABAergic marker *Slc32a1* encoding the vesicular inhibitory amino acid transporter was expressed at the second highest level in cluster 4 relative to all other clusters (*Figure 1—source data 1*), consistent with the GABAergic phenotype of mature AGRP/NPY neurons (*Figure 3—figure supplement 1*).

## Cluster 5

*Pomc*(low)*/Tac2* represents a distinct neuronal population from cluster 1 in its patterns of TFs associated with the development of mature POMC neurons and its low level of *Pomc* transcripts (*Figure 3E*). In particular, *Isl1* was highly expressed only at E11.5 and *Prdm12* at E11.5 and E13.5 in cluster 5. Starting from E15.5 through P12 there was a gradual onset of expression for a set of genes including *Tac2, Ar, Esr1, Tacr3, Prlr, Pdyn,* and *Kiss1/Gm28040* isoforms (*Figure 3E*). Confirmations of co-expression for *Pomc* with either *Tac2, Tacr3* and *Sox14* or TDTOMATO with ESR1 at several embryonic ages are shown in *Figure 3F*. Interestingly, *Sox14* has been reported to be necessary for expression of *Kiss1* (*Huisman et al., 2019*). Most of the top enriched genes for this cluster are inferred to be expressed at the middle to end portions of the pseudotime prediction, and their functions are related to reproduction, synapse assembly/organization, pre/post-synaptic specialization, and neuron migration (*Figure 3G*). Taken together, this ensemble of genes is characteristic of adult Arc KNDY neurons that play an essential role in the control of reproductive physiology. A previous publication based on transgene lineage tracing also concluded that a significant subpopulation of KNDY neurons is derived from POMC progenitor cells (*Sanz et al., 2015*).

A prominent and unique characteristic of cluster 6. *Pomc*(low)*/Dlx5* neurons was the expression of members of the *Dlx* family of homeobox TFs (*Figure 3—figure supplement 2A and B*). The six family members are usually expressed in three linked pairs of isoforms: 1 and 2, 3 and 4, or 5 and 6. Cluster 6 was enriched in *Dlx1/2*, and *Dlx5/6* together with *Dlxos1*. The *Dlx1/2* pair is known to activate expression of *Ghrh* in the mouse Arc (*Lee et al., 2018*). However, there were no transcripts for *Ghrh* expressed in cluster 6 or any of the other *Pomc* neuronal clusters suggesting that *Dlx1/2* are not sufficient to induce *Ghrh* expression. Moreover, it has been recently shown that *Dlx1/2* binding to *Otp* regulatory elements inhibits OTP production, subsequently reducing downstream *Agrp* expression (*Lee et al., 2018*). Cluster 6 had the highest expression of the opioid prepronociceptin (*Pnoc*) compared to the other integrated clusters, but low levels of preprodynorphin (*Pdyn*), similar to cluster 5 KNDY neurons.

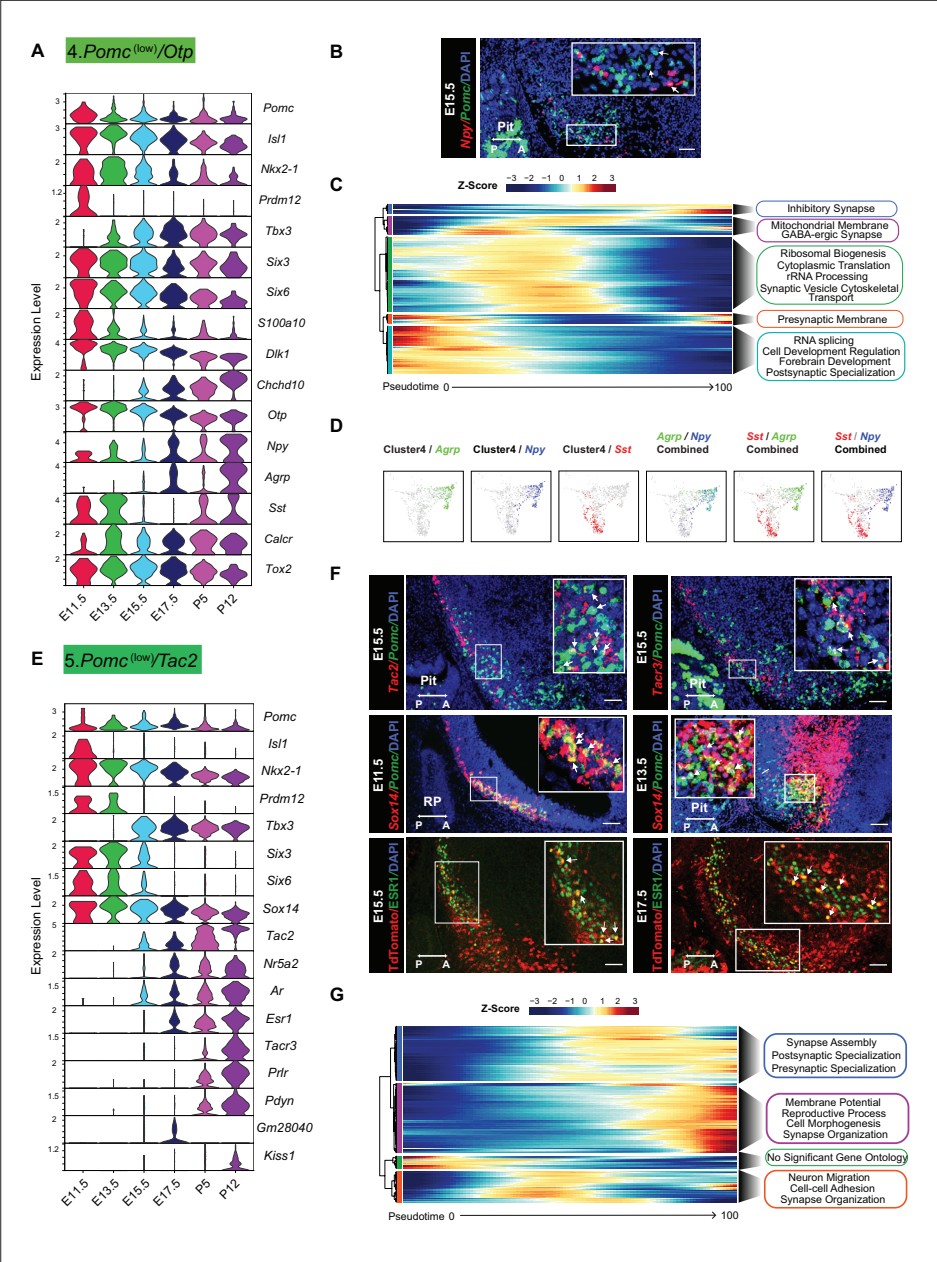

**Figure 3.** Temporal gene expression patterns of the *Pomc*(low)/*Otp* cluster and *Pomc*(low)/*Tac2* cluster. (**A** and **E**) Violin plots showing the expression of signature genes at each developmental stage in *Pomc*(low)/*Otp* cluster and *Pomc*(low)/*Tac2* cluster, respectively. (**B**) Fluorescence in situ hybridization showing the co-localization of *Pomc* (green) and *Npy* (red) at E15.5. (**C** and **G**) Heatmaps showing the gene ontology analysis based on the top marker genes in the order of pseudotime in *Pomc*(low)/*Otp* cluster and *Pomc*(low)/*Tac2* cluster, respectively. (**D**) Uniform Manifold Approximation and Projection (UMAP) plots showing cells in the *Pomc*(low)/*Otp* cluster expressing *Sst*, *Agrp*, *Npy*, *Sst*/*Agrp*, *Sst*/ *Npy*, or *Agrp*/*Npy*. Cells expressing *Sst* and cells expressing both *Agrp*/*Npy* are from two distinct cell populations. (**F**) Fluorescence in situ hybridization showing the co-localization of *Pomc* (green) and *Tac2* (red), the co-localization of *Pomc* (green) and *Tacr3* (red), and the co-localization of *Pomc* (green) and *Sox14* (red) at the indicated developmental stages. Immunofluorescence showing the co-localization of TDTOMATO (red) and ESR1 (green) at the indicated developmental stages. Insets are magnified views of the indicated boxes. Arrows indicate co-expressing neurons in the merged panels. Scale bar: 50 μm. Image orientation: left, posterior; right, anterior; Pit: pituitary gland, RP: Rathke's pouch.

The online version of this article includes the following figure supplement(s) for figure 3:

**Figure supplement 1.** Distribution of GABAergic and glutamatergic neurons in POMC(DsRed) cell clusters.

*Figure 3 continued on next page*

*Figure 3 continued*

**Figure supplement 2.** Temporal gene expression patterns of the *Pomc*[(low)]/*Dlx5* and *Pomc*[(low)]/*Prdm13* clusters.

## Cluster 6

It had the highest *Gad1*, *Gad2*, and *Slc32a1* (*Vgat*, vesicular inhibitory amino acid transporter) expression density of all clusters with virtually no *Slc17a6* (*Vglut2*, vesicular glutamate transporter 2) expression, suggesting that these differentiating neurons were exclusively GABAergic. A comparison of GABAergic and glutamatergic markers among the eight neuronal clusters is shown in *Figure 3—figure supplement 1*. Cluster 4 neurons were also uniformly GABAergic. Cluster 1 neurons contained a mixture of glutamatergic and GABAergic markers with the exception of undetectable *Slc32a1*. However, they did express low levels of *Slc18a2* (*Vmat2*, vesicular monoamine transporter 2), which possibly functions as an alternative vesicular GABA transporter in some dopamine neurons (*German et al., 2015*). This combination of mixed glutamatergic and GABAergic features in cluster 1 is characteristic of postnatal POMC neurons (*Wittmann et al., 2013*; *Jones et al., 2019*) and was also present to a limited extent in cluster 2. The remaining neuronal clusters 3, 5, 7, and 8 were all uniformly glutamatergic.

## Cluster 7

*Pomc*[(low)]/*Prdm13* contained the smallest number of cells among the eight neuronal clusters, and the cells were primarily limited to developmental ages E13.5–E17.5 (*Figure 1B*). Feature genes included *Prdm13*, *Nr5a1*, *Adcypap1*, *Cnr1*, *Fam19a1*, *Rbp1*, and *Pdyn* (*Figure 3—figure supplement 2C*). *Prdm13* was reported recently (*Chen et al., 2020*) to be highly expressed in E15.5 POMC cells.

We further analyzed all of the neurons and non-neuronal cells by reclustering them based on their individual developmental ages (*Figure 4*) rather than for their gene expression profiles integrated across all ages. Average gene expression levels for each of these developmental subclusters are listed in *Figure 4—source data 1* and the feature genes defining the developmental subclusters are listed in *Figure 4—source data 2*. Based on these data, it was then possible to trace the temporal continuity between developmental subclusters from age E11.5 to P12 relative to the original integrated cell cluster identities (*Figure 4—figure supplement 1*). These data are important because they connected the dynamic nature of gene expression profiles at each embryonic stage leading to the eventual transcriptomes present in the postnatal time points.

## Comparison of transcriptional profiles for neuropeptides, neuropeptide processing enzymes, neuroendocrine secretory proteins, and GPCRs in the eight neuronal POMC clusters at each developmental age

We identified 19 unique neuropeptide prohormone genes that were clearly differentially expressed features in at least one of the neuronal clusters at one or more developmental ages (*Figure 4—figure supplement 2*, *Figure 2—source data 1*). In addition to the neuropeptide genes already mentioned as characteristic features of certain clusters, additional neuropeptides were featured in other clusters primarily at postnatal ages P5 and P12. Notable examples are *Gal*, *Tac1*, *Adcyap1*, *Cartpt*, and *Pthlh*. Unlike the mosaic of neuropeptide gene expression profiles across clusters and developmental ages, the neuroendocrine secretory proteins, characteristic of dense core granules, were expressed ubiquitously in all eight neuronal clusters with similar gradients across all developmental ages (*Figure 4—figure supplement 2*). These included members of the chromogranin (*Chga* and *Chgb*) and secretogranin (*Scg2*, *Scg3*, and *Scg5*) families. Among the prohormone processing enzymes, *Pcsk1* was prominently expressed only at P5 and P12 in most neuronal clusters while *Pcsk2* was expressed at both embryonic and postnatal ages in the same clusters as *Pcsk1*. *Pam* was expressed in the same cluster/developmental age pattern as *Pcsk1*. Only *Cpe* was expressed in virtually all neurons at all developmental ages.

Split plots of G-protein-coupled receptor (GPCR) gene expression revealed a wide range of cluster- and age-specific patterns (*Figure 4—figure supplement 3*). *Cnr1* (cannabinoid receptor 1) was an important feature primarily of cluster 7 and the two interrelated clusters 3 and 8. The two metabotropic GABA receptors *Gabbr1* and *Gabbr2* were strongly expressed in all eight neuronal clusters but

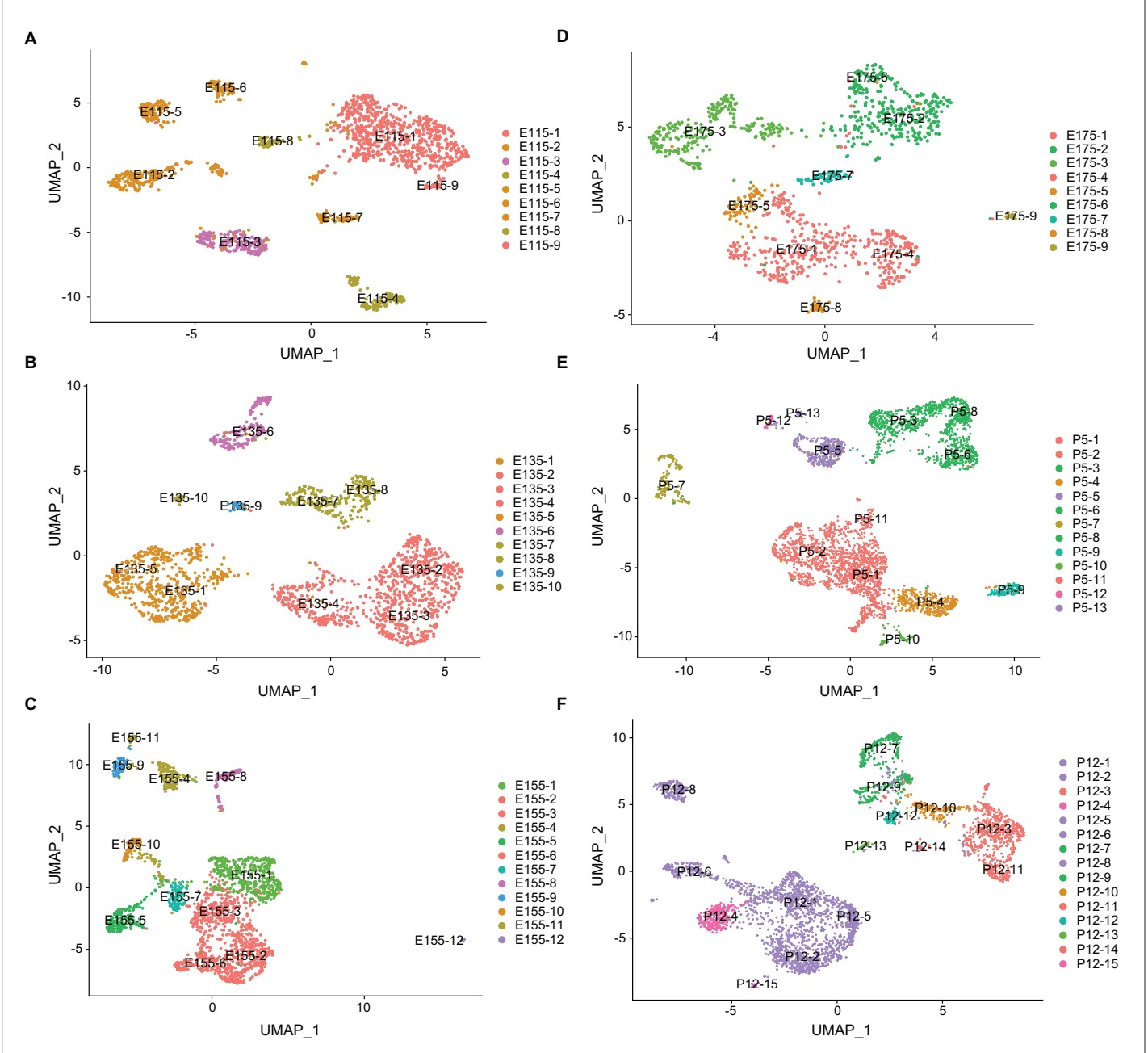

**Figure 4.** Uniform Manifold Approximation and Projection (UMAP) plots showing the distribution of major cell clusters at each developmental stage. (**A**) E11.5, (**B**) E13.5, (**C**) E15.5, (**D**) E17.5, (**E**) P5, and (**F**) P12. The developmental subclusters derived from each primary cluster are colored to corresponding cells in the UMAP plot in *Figure 1B*.

The online version of this article includes the following source data and figure supplement(s) for figure 4:

**Source data 1.** Average gene expression obtained from the Seurat analysis of each subcluster unique to the six developmental stages (corresponding to *Figure 4* and *Figure 4—figure supplement 1*).

**Source data 2.** Feature genes that define each subcluster at each of the six developmental stages (corresponding to *Figure 4* and *Figure 4—figure supplement 1*).

**Figure supplement 1.** POMC^DsRed developmental cell subclusters shown in *Figure 4* are projected to the original integrated cell clusters from *Figure 1*.

**Figure supplement 2.** Overview of neuropeptide, secretory granule, and processing enzyme gene expression patterns across clusters and developmental ages.

**Figure supplement 3.** Overview of G-protein-coupled receptor (GPCR) gene expression patterns across clusters and developmental ages.

differed in their developmental gradients. *Gabbr1* was expressed at all developmental ages while *Gabbr2* transcripts were largely present only in more mature neurons at postnatal ages P5 and P12. The opioid receptor *Oprl1* that is selectively activated by nociception/orphanin FQ (*Toll et al., 2016*) stood out for its consistently strong expression in all eight neuronal clusters, particularly at postnatal days P5 and P12.

The prokineticin receptor 1 (*Prokr1*) was selectively expressed together with the GPCR accessory protein gene *Mrap2* in cluster 1 *Pomc*(high)/*Prdm12* neurons. Although MRAP2 was initially identified as an activating modulator of melanocortin receptor signaling, consequently reducing food intake, it was subsequently shown to be promiscuous in its interaction with additional GPCRs (*Srisai et al., 2017*). Unlike its action on MC4R signaling, MRAP2 inhibits PROKR1 signaling to promote food intake in mice independently of its interaction with MC4R (*Chaly et al., 2016*). Therefore, the co-expression of *Prokr1* and *Mrap2* in anorexigenic POMC neurons suggests an additional mechanism for PROKR1's regulation of energy homeostasis by modulating the release of melanocortins from POMC neurons. Finally, the high expression of *Npy2r* (neuropeptide Y receptor 2) in both clusters 1 and 4 matches well with its expression in adult anorexigenic POMC neurons and orexigenic AGRP/NPY neurons, respectively.

## TRAP-seq analysis of POMC-expressing cells at P12 and P60

To define the important genetic programs that direct the transition from early postnatal to adulthood and to compare the differences of the transcriptional programs that guide embryonic vs. postnatal/adult development, we performed Translating Ribosome Affinity Purification TRAP-seq using compound *Pomc-CreERT2; Rosa26-eGFPL10a* transgenic mice where POMC neurons are labeled with a *eGFPL10a* tag after tamoxifen administration at specific developmental stages. Colocalization of POMC and GFP immunoreactivity confirms the expression of *eGFPL10a* in Arc POMC neurons upon tamoxifen injection (*Figure 5A*). Anti-GFP-conjugated beads were used to pull down actively translating RNAs bound to the eGFPL10a ribosomes from hypothalamic extracts. Both the pull-down mRNA samples and the resultant supernatant mRNA samples were subjected to RNA sequencing. Principal component (PC) analysis shows distinct separation of pull-down and supernatant samples at both ages P12 and P60 from three independent experiments, suggesting good quality of these samples and the intrinsic differences between pull-down and supernatant samples (*Figure 5B*). Both *Pomc* and *GFP* were highly expressed in TRAP pull-down samples relative to the supernatants (*Figure 5C*), further validating the specificity of the TRAP-seq method. Gene enrichment analysis identified 1143 and 1047 highly enriched genes ($p < 0.05$) from *Pomc- eGFPL10a* P12 and P60 TRAP-seq pull-downs, respectively (*Figure 5D*, *Figure 5—source data 1* and *Figure 5—source data 2*), from which 653 genes expressed in both datasets (*Figure 5E*). We hypothesize that the commonly expressed genes may be associated with the maintenance of POMC functional identity, whereas the uniquely expressed genes at P12 or P60 may be related to distinct age-dependent biological phenomena. To test this hypothesis, we performed gene ontology analysis on the 653 co-expressed genes, 394 P60 specifically expressed genes and 490 P12 specifically expressed genes. The results demonstrate that the commonly expressed genes are associated with the establishment and maintenance of neuron structure and basic neuronal functions. The top three functional annotations include the organization of synapses, regulation of neuron differentiation, and cytoskeleton-dependent intracellular transport. The ontology annotations on the uniquely expressed genes from P12 and P60 are dramatically different from each other. The functions of P12 uniquely enriched genes emphasize basic cellular metabolism, cellular modeling, and biochemical changes, whereas, the P60 enriched genes are related to the maturation of POMC neurons such as neurotransmitter development and POMC-specific function establishment including the responses to nutrient, and the construction of feeding behavioral circuits (*Figure 5F*). We next compared the TRAP-seq commonly expressed genes to our developmental scRNA-seq data to examine their expression levels in each cluster and each age. Approximately one-third (241) of the genes shown in the heatmaps on the left are abundantly expressing in the scRNA-seq dataset across all the clusters. Notably, compared to the four embryonic stages, a constantly higher transcript abundance was observed at both P5 and P12 ages (gray outlined boxes in *Figure 5G*), indicating great genetic similarities between P5 and P12. Intriguingly, the patterns of expression levels are very similar in both the P12 and P60 TRAP-seq datasets (*Figure 5G*, right heatmaps). To gain a better visualization of the distribution of TRAP-seq common

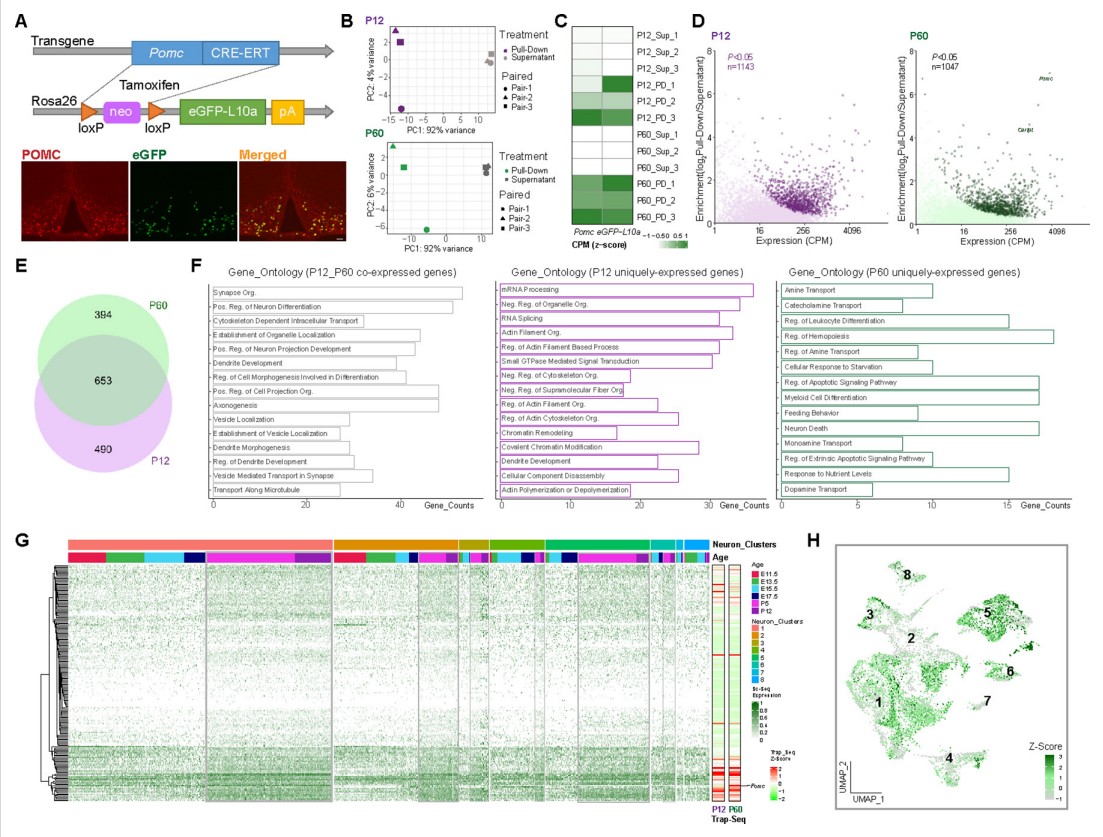

**Figure 5.** Translating ribosome affinity purification with RNA-sequencing (TRAP-seq) showing gene enrichment and gene expression profiles at P12 and P60. (**A**) Schematic diagram showing the generation of *Pomc-CreERT2; ROSA26^eGFP-L10a* mice for TRAP-seq experiment. Immunofluorescence validated the co-localization of proopiomelanocortin (POMC) and eGFP. (**B**) Principal component analysis showing the separation of the RNA sequencing data from beads pull-down vs. supernatant at P12 (purple) and P60 (green), respectively. (**C**) Heatmap showing both *Pomc* and eGFP were highly expressed in beads pull-down samples. Rows are the biological replicates of each sample. Sup: Supernatant samples; PD: pull-down samples. Data were presented as scaled counts per million (CPM). (**D**) Gene enrichment plots showing 1143 genes and 1047 genes were significantly expressed in beads pull-down samples at P12 and P60, respectively (p < 0.05). (**E**) Venn diagram showing the number of genes highly enriched in both P12 (purple) and P60 (green) beads pull-down samples. (**F**) Gene ontology analysis showing the top 15 biological processes that were represented in P12 and P60 co-expressed genes, P12 uniquely expressed genes and P60 uniquely expressed genes. (**G**) Expression profile of the top enriched genes from both P12 and P60 TRAP-seq datasets across eight neuronal clusters; gray boxes indicate the higher expression of these genes in the two postnatal stages. The two heatmaps (right) indicating the top genes expression in P12 and P60 TRAP-seq datasets. (**H**) Uniform Manifold Approximation and Projection (UMAP) plot showing the distribution of the top enriched genes from both P12 and P60 TRAP-seq datasets in eight neuronal cell clusters. CPM: counts per million; org.: organization; reg.: regulation; pos.:positive; neg.: negative.

The online version of this article includes the following source data for figure 5:

**Source data 1.** Differentially expressed genes by RNA translating ribosome affinity purification with RNA-sequencing (TRAP-seq) between pull-down vs. supernatant at age P12 (corresponding to *Figure 5*).

**Source data 2.** Differentially expressed genes by RNA translating ribosome affinity purification with RNA-sequencing (TRAP-seq) between pull-down vs. supernatant at age P60 (corresponding to *Figure 5*).

genes expression at a single-cell level, we grouped the 241 genes as one module, calculated module scores based on the normalized counts and projected the scores to a UMAP plot of the eight neuronal clusters (*Figure 5H*). Most cells from cluster 1 (*Pomc*^(high)^/*Prdm12*), cluster 5 (*Pomc*^(low)^/*Tac2*), cluster 3 (*Pomc*^(med)^/*Nr4a2*), and a subcluster of cluster 2 (*Pomc*^(med)^/*Ebf1*) present a higher module score (high gene expression levels). Assessment together with *Figure 6B* confirms that most of the high scoring cells are derived from the two postnatal stages (P5 and P12).

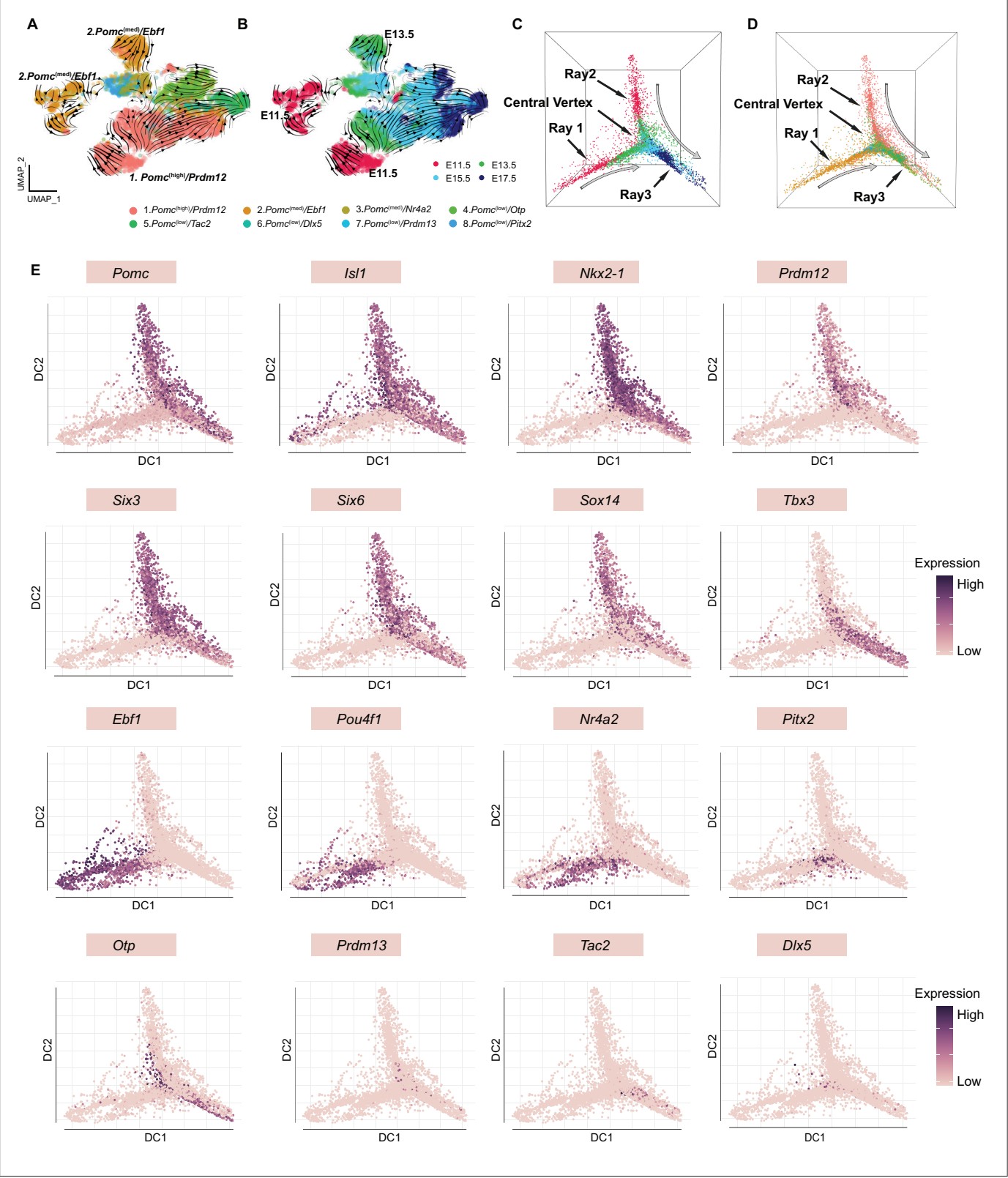

**Figure 6.** RNA velocity and diffusion maps analyses illustrating the developmental trajectories of hypothalamic neuronal clusters. (**A**) RNA velocity analysis showing the multiple origins of embryonic proopiomelanocortin (POMC) cells in eight neuronal clusters according to Seurat analysis cell embedding. (**B**) RNA velocity analysis showing the multiple origins of POMC cells correspond to the early embryonic stages. (**C–D**) Diffusion map showing the cell lineages of POMC neurons during embryogenesis. Cells are colored on the basis of their developmental stage (**C**) or based on their

*Figure 6 continued*

previously defined clusters (**D**). (**E**) Diffusion maps showing the expression of selected genes according to previous characterization of the feature genes representing each cluster. The color intensity corresponds to the normalized gene expression from Seurat analysis.

The online version of this article includes the following figure supplement(s) for figure 6:

**Figure supplement 1.** Overview of neuronal clusters split by developmental stages.

## Developmental relationships of gene expression patterns in the eight neuronal POMC clusters at four embryonic ages

To investigate the gene expression trajectories of developing POMC neurons, we performed an RNA velocity analysis of cells from the eight neuronal clusters focusing on the four embryonic stages. This RNA velocity analysis revealed a major origin of POMC neurons captured by the *Pomc*<sup>(high)</sup>/*Prdm12* cluster and a minor origin depicted by the *Pomc*<sup>(med)</sup>/*Ebf1* cluster (**Figure 6A and B**), which is consistent with the early appearance of these two clusters at E11.5 (**Figure 6—figure supplement 1**). Further investigation of cells from E13.5, E15.5, and E17.5 also suggest a dual origin from *Pomc*<sup>(high)</sup>/*Prdm12* or *Pomc*<sup>(med)</sup>/*Ebf1* clusters (**Figure 6—figure supplement 1**).

To further validate the hypothesis of a dual origins of POMC neurons and to better visualize cell trajectories in three-dimensional Euclidean space, we constructed diffusion maps of all neurons from the four embryonic stages (**Figure 6C**). The image depicts a triangular beveled star polygon with rays 1 and 2 derived primarily from distinct groups of neurons at ages E11.5 and E13.5, a center vertex containing individual neurons from the intermediate ages E13.5 and E15.5 and a third ray with neurons from ages E15.5 and E17.5. The same data are alternatively color-coded for each neuron's cluster identity (**Figure 6D**) showing that ray 1 is primarily derived from cluster 2, ray 2 from cluster 1, the center vertex from a convergence of all clusters 1–8 and ray 3 from clusters 1, 2, 4, 5, and 6. Compared to ray 2, cells in ray 1 are not continuously ordered, but instead show an abrupt transition at the junction of cells from E11.5 and E13.5.

We then overlaid the geometric location of each neuron with the expression levels of selected feature genes from each of the eight clusters (**Figure 6E**). Consistent with the diffusion maps analyses, the more granular data for individual genes and neurons illustrates the essentially identical patterns of high *Pomc* expression with the two originally discovered cognate transactivating factors *Isl1* and *Nkx2-1* across embryonic development (rays 2 and 3). However, *Isl1* was also expressed in numerous *Pomc*<sup>(low)</sup> neurons at E11.5 and E13.5 (ray 1). *Prdm12* and *Tbx3* exhibited complementary temporal patterns of high expression at early (ray 2) vs. late (ray 3) embryonic ages, respectively. The combined developmental and geometric patterns of *Six3, Six6,* and *Sox14* expression were all very similar to both *Pomc* and *Nkx2-1*, suggesting that all four of these TF genes might be necessary to define the cell fate of a subpopulation of hypothalamic neuronal progenitors into mature differentiated POMC neurons.

In contrast to those genes with expression patterns that defined the continuum of neuronal identity from ray 2 to ray 3, another set of genes including *Ebf1, Pou4f1,* and *Nr4a2* was similarly highly expressed only at the two earliest embryonic ages in neurons constituting ray 1. The expression of *Pitx2*, which characterizes cluster 8 and a portion of cluster 3, was high in ray 1 as it merges into the center vertex and was mostly derived from age E13.5. Finally, the temporal and geometric patterns of high expression levels for *Otp, Prdm13, Tac2,* and *Dlx5* were distinct from each other and the aforementioned genes, consistent with their ontogeny into the distinct differentiated cell fates of cluster 4. AGRP/NPY/GABA, cluster 5. KNDY, and cluster 6. GABA/PNOC neurons, respectively.

## Discussion

In this study, we tracked the genetic programs that give rise to *Pomc*-expressing neurons in the developing hypothalamus and followed their progression into terminally differentiated POMC neurons and alternative phenotypic destinies. To this end, we performed an scRNA-seq study specifically of *Pomc*-expressing neurons present in developing mouse hypothalami at four embryonic time points (E11.5, E13.5, E15.5, E17.5) and two early postnatal days (P5 and P12). We also performed a complementary TRAP-seq study on affinity purified translating RNAs derived from hypothalamic POMC cells obtained at ages P12 and P60. These developmental stages were selected based on the ontogeny of *Pomc*

expression in the presumptive hypothalamus (*Japón et al., 1994*), peak of neurogenesis of POMC neurons (*Nasif et al., 2015*; *Orquera et al., 2019*), subsequent developmental maturation (*Padilla et al., 2010*; *Sanz et al., 2015*), and terminal differentiation (*Quarta et al., 2019*).

Our study revealed the existence of a major group of POMC neurons (cluster 1. *Pomc*(high)/*Prdm12*) that showed by far the highest levels of *Pomc* UMIs relative to all other neuronal clusters at all ages. We believe that this cluster constitutes the canonical POMC neurons that further mature and integrate into functional circuits in the postnatal hypothalamus. Cluster 1 is featured by a unique combinatorial set of TFs that includes *Isl1*, *Nkx2-1,* and *Prdm12*. In the last few years, our laboratories demonstrated that these three TFs are present in neurons that start expressing *Pomc* in the developing arcuate nucleus at E10.5, and their expression continues at later developmental ages and adulthood in hypothalamic POMC neurons. Indeed, we showed that ISL1 (*Nasif et al., 2015*), NKX2-1 (*Orquera et al., 2019*), and PRDM12 (*Hael et al., 2020*) are absolutely necessary to specify the identity of POMC neurons. Moreover, we also demonstrated previously that adult mice lacking any of these three TFs exclusively from POMC cells display greatly reduced levels of hypothalamic *Pomc* mRNA together with increased adiposity and body weight. However, our results also indicated that the sole presence of ISL1, NKX2-1, and PRDM12 at E10.5–E11.5 is insufficient to determine the early identity of hypothalamic POMC neurons, suggesting that the combinatorial set of TFs necessary for neuronal-specific *Pomc* expression has additional early components, yet to be discovered. In this regard, our scRNA-seq study revealed several novel candidates in cluster 1 cells including *Six3*, *Six6*, *Sox1/2/3*, *Sox14,* and other TFs (*Figure 2*). Future molecular and functional genetic studies will be needed to assess their potential contribution to Arc *Pomc* expression and the functioning of POMC neurons.

*Six3* and *Six6* are two evolutionarily related TF genes known to play critical roles in the development of the eyes, forebrain, and pituitary (*Liu and Cvekl, 2017*; *Liu et al., 2010*; *Suh et al., 2010*). The expression patterns of *Six3* and *Six6* highly overlap during early embryogenesis but segregate into different territories during late embryonic development, indicating possible distinct postnatal functions (*Geng et al., 2008*). Indeed, *Six3* knockout mice die at birth whereas mice lacking *Six6* exhibit decreased numbers of GNRH neurons and impaired fertility (*Larder et al., 2011*). Although *Six3* and *Six6* are abundantly expressed in Arc POMC neurons across all examined developmental stages (*Figure 2*), their participation in the control of *Pomc* expression and the patterning and function of POMC neurons requires further investigation. Regarding Sox genes, a family of TFs known to regulate cell fate by transactivating cell-specific genes, in silico analysis of conserved SOX protein binding motifs suggests the potential binding of SOX proteins including SOX1, SOX2, SOX3, and SOX14 to *Pomc* neuronal enhancers nPE1 and nPE2 (*Chen et al., 2020*). All four of these *Sox* genes were highly enriched in *Pomc*(high)/*Prdm12* cluster 1 (*Figure 2* and *Figure 1—source data 2*) as well as in *Pomc*(low)/*Tac2* cluster 5 and *Pomc*(low)/*Prdm13* cluster 7. Furthermore, mice lacking *Sox14* exhibit a great reduction in KNDY neurons (*Huisman et al., 2019*).

A particular feature gene present in cluster 1 was *Tbx3*, a T-box TF that has been recently reported to play a key role in Arc *Pomc* expression (*Quarta et al., 2019*). Unlike *Isl1*, *Nkx2.1*, and *Prdm12*, which are all strictly required at the onset of *Pomc* expression (E10.5), *Tbx3* expression commences several days later (E15.5) in the Arc and, therefore, apparently does not participate in the early specification of POMC neurons. However, *Tbx3* plays an important role in the functional maturation of these neurons as a transcriptional booster of arcuate *Pomc* expression. It was recently demonstrated in mice specifically lacking *Tbx3* from POMC neurons that *Pomc* expression was diminished leading to hyperphagia and obesity (*Quarta et al., 2019*).

The TRAP-seq results add another layer to our understanding of the genetic machinery that leads to mature functional POMC neurons. Genes that are uniquely enriched at P12 are associated with the basic cellular biology process and cell metabolism whereas genes specifically expressed at P60 are involved in the establishment of POMC neuronal functions. The co-expressed genes from TRAP-seq are highly expressed at P5 and P12 in the single-cell sequencing datasets, suggesting that the genetic programs that dominate embryonic POMC neuronal development are substantially different from the programs that guide postnatal POMC neuronal differentiation. Although only 1 week apart, the cells at P5 share more genetic similarities to the cells at P12 rather than the cells at E17.5.

An unanticipated finding of our study is the discovery of an early population of Arc *Pomc*-expressing neurons characterized by the expression of the TF *Ebf1* and *Pou4f1*, in the absence of *Prdm12* transcripts. The expression profile of cluster 2. *Pomc*(med)/*Ebf1* differs from cluster 1. *Pomc*(high)/*Prdm12*

most particularly at the earliest developmental ages E11.5–E15.5 and may both represent alternative genetic routes leading to mature arcuate neurons. The early expression of *Ebf1* suggests that this TF may regulate the migration of differentiating neurons out of the ventricular zone to the mantle, as has been reported in the developing striatum (*Garel et al., 1999*). Further studies are needed to understand the functional significance of the non-canonical population of *Pomc*-expressing neurons captured by cluster 2 cells.

Another main finding of our study is the identification of multiple subpopulations of *Pomc*-expressing neurons that originate in the developing arcuate nucleus. In addition to the previously characterized fertility-regulating KNDY neurons (*Pomc*(low)/*Tac2*) (*Sanz et al., 2015*) and orexigenic AGRP/NPY neurons (*Pomc*(low)/*Otp*) (*Padilla et al., 2010*), our data indicated that, as early as E11.5, *Pomc*-expressing neurons may also give rise to four other less characterized neuronal populations noted as *Pomc*(med)/*Nr4a2*, *Pomc*(low)/*Pitx2*, *Pomc*(low)/*Dlx5,* and *Pomc*(low)/*Prdm13*. Moreover, subclustering cells from E11.5 revealed a distinct cell population featured by *Otp* expression (E11.5–9 cluster in *Figure 4*), suggesting that the presence of *Otp* transcripts at E11.5 may already be dictating the differentiation pathway of this neuronal cluster toward an AGRP phenotype, a hypothesis that requires further investigation. In addition, RNA-velocity developmental trajectory analysis (*Figure 6A* and *Figure 6—figure supplement 1B*) suggested that the early cells in *Pomc*(high)/*Prdm12* cluster project to *Pomc*(low)/*Otp* cluster.

*Pomc*(med)/*Nr4a2* and *Pomc*(low)/*Pitx2* are related clusters sharing 20% of their top 150 feature genes being one of the most salient differences the presence of neuropeptide *Cck* transcripts only in the *Pomc*(low)/*Pitx2* cluster. Comparing our dataset with a previous one obtained using FACS from adult hypothalami of POMC-EGFP mice (*Lam et al., 2017*) in which *Nr4a2* showed to be expressed in 18.5% of cells in cluster 4, it is likely that embryonic *Pomc*(med)/*Nr4a2* cells captured by our study keep their identity through adulthood. Similarly, signature genes for the *Pomc*(low)/*Dlx5* cluster including *Dlx1, Dlx2, Dlx5,* and *Dlx6* were also found to be highly enriched and specific for cluster 1 in Lam et al.'s study, suggesting that the *Pomc*(low)/*Dlx5* cluster detected in our study becomes a subset of adult POMC neurons. Finally, our identification of the *Pomc*(low)/*Prdm13* cluster matches with a recent study showing that *Prdm13* is highly enriched in FACS cells from E15.5 POMC-EGFP embryos analyzed by whole transcriptome RNA sequencing (*Chen et al., 2020*).

It is worth mentioning that despite some extent of similarities between embryonic and adult POMC cells, the majority of embryonic and early postnatal cells profiled in our study are distantly related to mature adult POMC neurons, indicating that terminal differentiation of POMC neurons is a long-lasting maturation process that proceeds along multiple life stages. For example, most adult POMC cells express proprotein convertase PC1/3 (*Pcsk1*) and leptin receptor (*Lepr*), whereas only 22% and 2.7% of cells from our datasets express the two genes, respectively, suggesting critical functional changes in the transcriptome of POMC neurons during postnatal maturation after early postnatal ages.

Due to the limitations of the scRNA-seq techniques, some interesting questions cannot be answered from our study at the functional level. For example, are all TFs present in cluster 1. *Pomc*(high)/*Prdm12* at E11.5 critical for dictating the identity of the cannonical POMC neurons? what TFs are critical for maintaining POMC neuron identity throughout lifetime? Is *Otp* the only critical gene that determines the transition of *Pomc*-expressing cells into AGRP/NPY neurons? Which are the TFs that give rise to the lineage of KNDY neurons during development? We believe that our study provides a valuable resource to tackle these and many other questions aimed at a better understanding of the regulation and functioning of POMC and other related neurons in the mammalian arcuate nucleus of the hypothalamus.

## Concluding remarks

In summary, this study is the first to comprehensively characterize the transcriptomes of Arc hypothalamic POMC cells during embryonic and early postnatal development. This dataset extends our understanding of the diversification of early POMC cells and provides a valuable resource for further elucidating the regulatory mechanism for *Pomc* expression and neuronal maturation. The identification of new marker genes in each subpopulation from scRNA-seq and uniquely expressed genes at P12 or P60 from the TRAP-seq study will potentially lead to new directions for future functional studies of POMC neurons.

# Materials and methods

**Key resources table**

| Reagent type (species) or resource | Designation | Source or reference | Identifiers | Additional information |
|---|---|---|---|---|
| Genetic reagent (*Mus musculus*) | *Pomc -TdDsRed* | PMID:19864580 | | Dr Malcolm Low (University of Michigan) |
| Genetic reagent (*Mus musculus*) | *Pomc-CreERT2* | PMID:24177424 | RRID: MGI:5569339 | |
| Genetic reagent (*Mus musculus*) | *Rosa26eGFP-L10a* | Jackson Laboratory | Stock # 024750; RRID: IMSR_JAX:024750 | |
| Antibody | Anti-Nr5a1(Rabbit Polyclonal) | Dr Gary Hammer, University of Michigan | | IF (1:750–1:1000) |
| Antibody | Anti-td-Tomato [16D7] (Rat Monoclonal) | Kerafast | Cat # EST203; RRID:AB_2732803 | IF (1:750) |
| Antibody | Anti-Pomc (Rabbit Polyclonal) | Phoenix Pharmaceuticals | Cat # H-029–30; RRID:AB_2307442 | IF(1:1000) |
| Antibody | Anti-Rabbit Alexa 488 (Goat polyclonal) | Invitrogen | Cat # A-11034; RRID:AB_2576217 | IF(1:500–1:1000) |
| Antibody | Anti-Rat Alexa Fluor 555 (Goat polyclonal) | Invitrogen | Cat # A-21434; RRID:AB_2535855 | IF(1:500–1:1000) |
| Antibody | Anti-Esr1 (Rabbit Polyclonal) | Dr Sue Moenter, University of Michigan | | IF(1:10000) |
| Antibody | Anti-GFP (Chicken polyclonal) | Abcam | Cat # ab13970; RRID:AB_300798 | IF(1:1000) |
| Chemical compound, drug | Papain Dissociation System | Worthington Biochemical Corporation | LK003178 | |
| Chemical compound, drug | DNase | Worthington Biochemical Corporation | LK003172 | |
| Software, algorithm | R | R Project for Statistical Computing | https://www.r-project.org/ | |
| Software, algorithm | Seurat 3.1.5 | Satija Lab | https://satijalab.org/seurat/articles/install.html | |
| Software, algorithm | Monocle 2 | Cole-trapnell lab | http://cole-trapnell-lab.github.io/monocle-release/docs/ | |
| Software, algorithm | Destiny 2.14.0 | Carsten Marr & Florian Buettner lab | https://theislab.github.io/destiny/ | |
| Sequence-based reagent | RNAscope Multiplex Fluorescent V2 Assay | ACDbio | Cat # 323110 | |
| Sequence-based reagent | Mm-*Six3* | Advanced Cell Diagnostics | Cat # 412941-C3 | |
| Sequence-based reagent | Mm-*Six6* | Advanced Cell Diagnostics | Cat # 574291 | |
| Sequence-based reagent | Mm-*Tac2* | Advanced Cell Diagnostics | Cat # 446391-C3 | |
| Sequence-based reagent | Mm-*Npy* | Advanced Cell Diagnostics | Cat # 313321 | |
| Sequence-based reagent | Mm-*Ebf1* | Advanced Cell Diagnostics | Cat # 433411 | |
| Sequence-based reagent | Mm-*Pou4f1* | Advanced Cell Diagnostics | Cat # 414671-C3 | |
| Sequence-based reagent | Mm-*Pomc* | Advanced Cell Diagnostics | Cat # 314081-C2 | |
| Sequence-based reagent | Mm-*Sox14* | Advanced Cell Diagnostics | Cat # 516411 | |
| Sequence-based reagent | Mm-*Prdm13* | Advanced Cell Diagnostics | Cat # 543551-C2 | |
| Sequence-based reagent | Mm-*Tacr3* | Advanced Cell Diagnostics | Cat # 481671 | |
| Sequence-based reagent | Mm-*Dlx5* | Advanced Cell Diagnostics | Cat # 478151 | |

*Continued on next page*

*Continued*

| Reagent type (species) or resource | Designation | Source or reference | Identifiers | Additional information |
|---|---|---|---|---|
| Sequence-based reagent | Mm-*Pitx2* | Advanced Cell Diagnostics | Cat # 412841 | |
| Sequence-based reagent | Mm-*Pomc* | Advanced Cell Diagnostics | Cat # 314081 | |
| Sequence-based reagent | Mm-Positive Control Probe | Advanced Cell Diagnostics | Cat # 320881 | |
| Sequence-based reagent | Mm-Negative Control Probe | Advanced Cell Diagnostics | Cat # 320871 | |
| Sequence-based reagent | RNAscope Probe Diluent | Advanced Cell Diagnostics | Cat # 300041 | |
| Chemical compound, drug | Opal 520 Reagent Pack | AKOYA Biosciences | Cat # NC1601877 | 1:750–1:1500 |
| Chemical compound, drug | Opal 620 Reagent Pack | AKOYA Biosciences | Cat # NC1612059 | 1:750–1:1500 |
| Chemical compound, drug | Opal 690 Reagent Pack | AKOYA Biosciences | Cat # NC1605064 | 1:750–1:1500 |
| Other | DAPI | BD Biosciences | Cat # 564907 | FACS (500–1000 ng/mL) |
| Other | ProLong Gold Antifade Mountant | ThermoFisher | P36930 | |
| Other | Earle's Balanced Salts | Sigma-Aldrich | E2888 | |

## Mice

All procedures were performed in accordance with the Institutional Animal Care and Use Committee (IACUC) at the University of Michigan and followed the Public Health Service guidelines for the humane care and use of experimental animals. Mice were housed in ventilated cages under controlled temperature and photoperiod (12 hr light/12 hr dark cycle, lights on from 6:00 AM to 6:00 PM), with free access to tap water and laboratory chow (5L0D, LabDiet). Breeding mice were fed with the breeder chow diet (5008, LabDiet). Transgenic mice expressing the fluorescent protein Tdimer-Discosoma red (TdDsRed) in POMC neurons were generated previously (*Hael et al., 2020*). The vast majority of *Pomc-TdDsRed* cells were validated as authentic POMC neurons based on co-localization of ACTH immunostaining with the TdDsRed fluorophore (*Hentges et al., 2009*) *Pomc-CreERT2* and *Rosa26*$^{eGFP-L10a}$ (Stock no. 024750, The Jackson Laboratory) mice were generated as previously described (*Liu et al., 2014*). To induce POMC neurons-specific expression of eGFP-L10a at P12 or P60, tamoxifen (50 mg/kg) was injected intraperitoneally for 5 constitutive days from P6 to P10 or P50 to P54.

## Generation of single-cell suspension for scRNA-seq

Adult male *Pomc-TdDsRed/+* mice were bred with female *Pomc-TdDsRed/+* mice and the day of copulation plug detection was counted as embryonic day 0.5 (E0.5). Embryonic hypothalami were dissected at E11.5, E13.5, E15.5, and E17.5. Postnatal hypothalami were isolated from pups at postnatal day 5 (P5) and P12. In each dissection, hypothalami from at least six pups were pooled together to acquire enough cells for FACS. Pooled tissue samples were digested in papain solution supplemented with DNAse for 20 min (E11.5 and E13.5), 30 min (E15.5), 40 min (E17.5), or 1 hr (P5 and P12) with gentle agitation based on a published protocol (*Garel et al., 1999*). The isolated cells were stained with DAPI as an indicator for cell viability and sorted by FACS for the transgenic red fluorophore. The collected cells were counted and subjected to scRNA-seq library preparation and sequencing (10× Genomics).

## scRNA-seq data processing

A total of 37,053 single cells (E11.5: 5329, E13.5: 7051, E15.5: 5148, E17.5: 2488, P5: 7551, and P12: 9486) from six different developmental stages were processed using the 10× Genomics Chromium system. The libraries were sequenced on Illumina HiSeq 4000 and NovaSeq platforms. We obtained a total of over 2.6 billion reads with an average of 70,7896 reads per cell. Over 85% of reads mapped confidently to the mouse genome across all six developmental stages. Raw reads were processed with

Cell Ranger (v.2.2 and v.3.0). Seurat package (v.3.1.5) (*Butler et al., 2018*) was used for downstream analysis. Since the number of genes and UMI counts varied across developmental stage, we applied different criteria at each stage to filter out possible doublets and low-quality cells. Specifically, at E11.5, we removed outlier cells that had UMI counts <5000 or >60,000 and gene counts <1000 or >8000 (determined by the visualization of UMI counts and gene count distributions). At E13.5, we removed outlier cells that had UMI counts <2500 or >30,000 and gene counts >1500. At E15.5, we removed outlier cells that had UMI counts >40,000 and gene counts <1000 or >6000. At E17.5, we removed outlier cells that had UMI counts >25,000 and gene counts <1000 or > 5000. At P5, we removed outlier cells that had UMI counts <1500 or >60,000 and gene counts <1500 or >8000. At P12, we removed outlier cells that had UMI counts <2000 or >60,000 and gene counts <1500 or >8000. Moreover, cells with high proportions of mitochondrial genes (>10%) or hemoglobin genes (>10%) were filtered out. Finally, we removed all the cells without any *Pomc* transcript UMI counts. A total of 13,953 cells passed the criteria (E11.5: 1498, E13.5: 1796, E15.5: 2078, E17.5: 1139, P5: 3909 and P12: 3533) for downstream analysis.

The gene expression level for each cell was normalized by the total expression, multiplied by a scaling factor of 10,000 followed by log-transformation (*Butler et al., 2018*). The top 2000 most variable genes were selected from each age and 60 integration anchors were used to combine all the datasets together. After scaling and centering the integrated dataset, we performed PCs analysis on the data matrix and the top 50 PCs were selected based on JackStraw and Elbow plots from Seurat package (*Hentges et al., 2009*) for data visualization using Uniform Manifold Approximation and Projection (UMAP) technique. We next constructed a shared nearest neighbor (SNN) graph by setting an expected number of neighbors to 50. In order to cluster the cells, an SNN modularity optimization technique within a function FindClusters was used to group cells together with a resolution parameter 0.1. The identity for each cluster was assigned based on the level of *Pomc* transcripts and prior knowledge of marker genes. The similar analysis was applied to define *TdDsRed* cell clusters (*Figure 1—figure supplement 2* and *Figure 1—figure supplement 3*). In addition to the above criteria applied to remove unwanted cells, we chose cells with at least one *TdDsRed* transcript UMI count and used the top 2000 highly variable genes for PCs analysis. The top 30 PCs were selected for unsupervised clustering with a resolution of 0.3.

Unsupervised cell clustering from each developmental age (*Figure 4*) was conducting using the following PC numbers (E11.5: 25; E13.5: 24, E15.5: 26, E17.5: 19, P5: 40, P12: 34) with their corresponding resolutions (E11.5: 0.2; E13.5: 0.6, E15.5: 0.5, E17.5: 0.5, P5: 0.5, P12: 0.5). The UMAP plots, violin plots, feature plots, and dot plots were all generated in the Seurat package. The average expression levels for each gene within each developmental subcluster (*Figure 4—source data 1*) were obtained from Seurat and calculated by the average of [*exp(logNormalized counts)–1*]. The differentially expressed genes in each identified cluster was identified by the comparison of gene expression levels in a specific cluster to all the other clusters.

## Cluster identity comparisons

To compare cluster identities between POMC cells and DSRED cells (*Figure 1—figure supplement 2*), we normalized the DSRED cells data and acquired the top 2000 most variable genes. We then use the standard datasets integration method from Seurat packages with the top 50 anchors for data integration and top 50 anchors to transfer cell identities from POMC cells to DSRED cells. We also applied this approach to conduct age-pairwise comparisons (*Figure 4—figure supplement 1*) with default setting from Seurat V3.1.5 package.

## Cell lineage construction

For RNA velocity analysis, spliced and unspliced matrices of reads were summarized using velocyto (v.0.17.16) with default parameters (*La Manno et al., 2018*). Low complexity and repeat regions were downloaded from the UCSC browser (rmsk table from mm10). scVelo was performed for RNA velocity analysis (v.0.2.2) (*Bergen et al., 2020*). The cell embedding information was acquired from Seurat analysis. The top 4000 genes, the top 20 PCs, and the top 20 neighbors under stochastic mode were used to generate velocity graph.

Cell lineage was constructed using the Destiny (v.2.14.0) package, which implements the formulation of diffusion maps (*Angerer et al., 2016*). Diffusion maps are a spectral method for non-linear

dimension reduction, which is especially suitable for analyzing single-cell gene expression data from different time-courses. We removed VLMC, glial cells, and astrocytes (clusters 9, 10, and 11 in *Figure 1*) and kept only neurons at embryonic stages (E11.5, E13.5, E15.5, and E17.5), resulting in a total of 6005 cells for this analysis. The log-transformed normalized data from Seurat data slot and the corresponding annotation information were imported to construct an expression matrix. The diffusion maps were constructed under the default setting for the Gaussian kernel width sigma ($\sigma$) and 300 nearest neighbors. The first two diffusion components (DCs) were used to visualize the results. The 3D plots were produced using rgl package (v.0.100.50) with the top three DCs.

To perform pseudotime gene expression analysis, cells from cluster 1, cluster 2, cluster 4, and cluster 5 were extracted from the Seurat object, respectively. Raw counts were acquired to construct Monocle (v.2.4.0) (*Qiu et al., 2017a*, *Trapnell et al., 2014*; *Qiu et al., 2017b*) datasets. Cells were ordered along the pseudotime by setting E11.5 as the root state. Differentially expressed genes (Log2FoldChange > 0.25 and p < 0.05) in each cluster from the Seurat analysis were plotted along the pseudotime using 'plot_pseudotime_heatmap' function. Gene ontology analysis (Biological Process) was performed using ClusterProfiler (v.3.18.1) (*Yu et al., 2012*).

## TRAP-seq analysis

Mice (*Pomc-CreERT2*; ROSA26$^{eGFP-L10a}$) were euthanized and decapitated. The brain was removed from the skull and the arcuate nucleus was isolated from 2 mm thick coronal slices using a brain matrix. Tissues were then homogenized and the subsequent tissue lysate was subjected to immunopurification steps according to the previous protocol (*Heiman et al., 2014*). The ribo-depletion kit (RiboGone, Takara, CA) and SMARTer Stranded total RNA sample prep kit were used to remove excess ribosomal RNA and synthesize cDNA library. Samples were sequenced on a 50-cycle single end run on a HiSeq 4000 (Illumina) according to manufacturer's protocols.

Raw sequencing data was processed at the University of Michigan Bioinformatics core. Briefly, the quality of the raw reads data was checked using FastQC (v.0.11.3) and the filtered reads were aligned to reference genome (UCSC mm10) using TopHat (v.2.0.13) and Bowtie2 (v.2.2.1) with default parameters. The HTSeq/DEseq2 method was used for differential expression analysis with paired-samples and treatment (pull-down vs. supernatant) as the main effects.

## Tissue collection and immunofluorescence staining

Depending on developmental stage, embryos at E10.5–E17.5 were fixed in 4% paraformaldehyde in phosphate buffer solution (PBS) at 4°C for a various period of time. Specifically, embryos at E10.5–E12.5 were fixed for 1 hr; heads from embryos at E15.5–E16.6 and E17.5 were fixed for 2 and 4 hr, respectively. Tissues were stabilized in 10% sucrose/10% gelatin in PBS at 37°C for 30 min prior to embedding in OCT compound as described previously (*Butler et al., 2018*). Embryonic brains were sectioned sagittally on a cryostat (Leica CM1950) at 20 µm thickness. For dual immunofluorescence, after washing excess OCT compound with PBS, a heat-induced antigen retrieval process was performed using citrate buffer (10 mM anhydrous citric acid and 0.05% Tween-20, pH 6.0) at 80°C for 30 min followed by two washes in PBS. Sections were then blocked with 3% normal goat serum and 0.1% Triton X-100 for 1 hr and incubated with primary antibodies at room temperature overnight. After PBS washes, sections were incubated with goat secondary antibodies for 2 hr at room temperature. Nuclei were stained with DAPI (1 mg/L) for 10 min and the slides washed five times with PBS before mounting with ProLong Gold Antifade Mountant (ThermoFisher). Sections were imaged on a Nikon 90i fluorescence microscope with NIS-Elements software. Information on the sources and dilution of antibodies are listed in Key resources table.

## In situ hybridization

FISH was performed using RNAscope Multiplex Fluorescent V2 Assay (Advanced Cell Diagnostics) according to the manufacturer's instructions with slight modifications. Specifically, after washing with PBS, sections were post-fixed with 4% paraformaldehyde for 20 min, and dehydrated in a series of ethanol solutions (50%, 70%, and 100%). After drying, dehydrated sections were incubated in RNAscope Hydrogen Peroxide solution for 10 min followed by the protease treatment using RNAscope Protease IV for 20 min. Sections were then hybridized to target RNA probes for 2 hr in the HybEZ II hybridization oven. Hybridized probes were amplified using a cascade of signal amplification

solutions (AMP 1–3) followed by standard signal developing protocol as described in the manufacturer's brochure. Detailed information on RNA probes and dilutions are listed in Key resources table. Confocal images were obtained using a Nikon Instruments A1 Confocal Laser Microscope with NIS-Elements software.

## Acknowledgements

We thank Dr Graham L Jones, Dr Zoe Thompson, and Charles Keane for their assistance in animal dissection, tissue sectioning, and data analysis, Dr Joshua Welch and Tongyu Liu for advice on data analysis, the Flow Cytometry Core, the Advanced Genomics Core, the Bioinformatics Core and the Microscopy, Imaging and Cellular Physiology Core (supported by NIH Grant: P30DK020572) and the University of Michigan Center for Gastrointestinal Research (supported by NIH grant: P30DK034933) at University of Michigan for data collection. Funding: This study was supported by NIH grant DK068400 (MJL and MR)

## Additional information

### Funding

| Funder | Grant reference number | Author |
| --- | --- | --- |
| National Institute of Diabetes and Digestive and Kidney Diseases | DK068400 | Malcolm J Low Marcelo Rubinstein |

The funders had no role in study design, data collection and interpretation, or the decision to submit the work for publication.

### Author contributions

Hui Yu, Conceptualization, Data curation, Formal analysis, Investigation, Methodology, Resources, Software, Validation, Visualization, Writing - original draft, Writing - review and editing; Marcelo Rubinstein, Conceptualization, Formal analysis, Funding acquisition, Methodology, Supervision, Writing - original draft, Writing - review and editing; Malcolm J Low, Conceptualization, Data curation, Formal analysis, Funding acquisition, Methodology, Resources, Supervision, Writing - original draft, Writing - review and editing

### Author ORCIDs

Hui Yu http://orcid.org/0000-0001-5249-0193
Marcelo Rubinstein http://orcid.org/0000-0002-7500-7771
Malcolm J Low http://orcid.org/0000-0002-9900-3708

### Ethics

All procedures were performed in accordance with the Institutional Animal Care and Use Committee (IACUC) protocol (PRO00008570) at the University of Michigan and followed the Public Health Service guidelines for the humane care and use of experimental animals.

### Decision letter and Author response

Decision letter https://doi.org/10.7554/eLife.72883.sa1
Author response https://doi.org/10.7554/eLife.72883.sa2

## Additional files

### Supplementary files

• Transparent reporting form

### Data availability

All raw data have been deposited in the Gene Expression Omnibus under accession numbers GSE154153 and GSE181539.

The following dataset was generated:

| Author(s) | Year | Dataset title | Dataset URL | Database and Identifier |
|---|---|---|---|---|
| Hui Y, Marcelo R, Malcolm L | 2021 | Single cell RNA-seq of the developing mouse hypothalamus | https://www.ncbi.nlm.nih.gov/geo/query/acc.cgi?acc=GSE154153 | NCBI Gene Expression Omnibus, GSE154153 |
| Hui Y, Marcelo R, Malcolm L | 2021 | Developmental single-cell transcriptomics of hypothalamic POMC progenitors reveal the genetic trajectories of multiple neuropeptidergic phenotypes | https://www.ncbi.nlm.nih.gov/geo/query/acc.cgi?acc=GSE181539 | NCBI Gene Expression Omnibus, GSE181539 |

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
