## [Editor Report]

This study investigates the developmental origins of functionally distinct neuronal populations in the arcuate nucleus of the hypothalamus. This study reports the transcriptional program of develop of multiple subtypes of POMC neurons, which are important for multiple physiological and behavioral functions, including appetite. The paper uses cutting edge methods will be important for research studying the arcuate nucleus including regulating food intake and metabolism.

---

## [Decision Letter]

**Decision letter after peer review:**

Thank you for submitting your article "Developmental single-cell transcriptomics of hypothalamic POMC progenitors reveal the genetic trajectories of multiple neuropeptidergic phenotypes" for consideration by *eLife*. Your article has been reviewed by 3 peer reviewers, one of whom is a member of our Board of Reviewing Editors, and the evaluation has been overseen by Marianne Bronner as the Senior Editor. The reviewers have opted to remain anonymous.

Essential revisions:

1) Showing in situ hybridization data for E11.5 markers in a coronal plane, where the ventricular and mantle layer can de distinguished, would help to resolve the question of whether Ebf-hi expression represents newly differentiated neurons that are in the process of migrating out of the ventricular zone. (This appears to be case from the E13.5 section in Figure 1F, but it is hard to tell.)

2) Addition of scRNA-seq data from earlier embryonic stages (i.e. E10.5 and/or E11.0) would help to identify the precursor to the population of immature POMC neurons that is captured by cluster 1.POMC-hi/Prdm12.

3) One question raised by these data are the presence of an inhibitory population of GABA releasing (ie Vgat-expressing) POMC neurons. This is controversial with some studies reporting this and others indicating that there are no GABAergic POMC neurons. Given the evidence for a developmental cluster that is evident even at P12, it is important to resolve this issue in the context of the authors report of a bona fide transcriptional cluster with these co-expression patterns. The authors should examine the colocalization of these Pomc/Vgat neurons using RNA-FISH and report their distribution across development.

4) Finally, several conclusions need to be tempered. Nonetheless, the paper will be an outstanding resource for the field. The discussion should be written to reflect the important resource of this data set.

*Reviewer #1 (Recommendations for the authors):*

There seemed to be a substantial subset of microglia expressing Pomc (Figure 1), did the authors remove these from the dataset? How do they determine a bona fied Pomc neuron vs a neuron with Pomc mRNA contamination.

A supplemental model figure/cartoon integrating the various developmental trajectories with their defining transcriptional factors would be useful to visualize the key findings of the paper.

*Reviewer #2 (Recommendations for the authors):*

• Showing in situ hybridization data for E11.5 markers in a coronal plane, where the ventricular and mantle layer can de distinguished, would help to resolve the question of whether Ebf-hi expression represents newly differentiated neurons that are in the process of migrating out of the ventricular zone. (This appears to be case from the E13.5 section in Figure 1F, but it is hard to tell.)

• Addition of scRNA-seq data from earlier embryonic stages (i.e. E10.5 and/or E11.0) would help to identify the precursor to the population of immature POMC neurons that is captured by cluster 1.POMC-hi/Prdm12.

• The number of total neurons changes dramatically across gestation, in large part because cells that adopt non-POMC fates turn off Pomc expression and are no longer captured by FACS. In Figure 1, reporting the absolute number of neurons in each cluster, instead expressing it as a percentage, would make it easier to interpret.

• Analyses shown in Figure 3 Supplemental Figures 3-4 are critical to resolve issues related to POMC progenitor heterogeneity. This information should be moved to the main figures. Maintaining a consistent color scheme between Figure 1 and Figure 3 Supplement 3 would make it much easier to interpret this data. The chart in Supplemental Figure 4 could then remain as a supplemental figure to reinforce this point.

• Trajectory analyses shown in Figure 5 should include separate analyses of 2.Pomc-med/Ebf1 and 3.Pomc-med/Nra2, as they appear to map onto different populations in the mature ARH (Supplemental Figure 15)

• There are discrepancies between the text (lines 489-497) and Supplementary file 14, which do not link 3.Pomc-med/Nr4a2 neurons to mature POMC populations and Supplementary file 15, which does, making it difficult to interpret these data.

*Reviewer #3 (Recommendations for the authors):*

This manuscript by Yu et al. beautifully captures the transcriptional heterogeneity of mouse POMC neurons across hypothalamic development. This study unifies multiple other observations about the role for other neuron al cell types that express POMC transiently during development. The paper is an important contribution to understanding of the diversity of POMC neuron classes and their relationship to other cell types in the arcuate nucleus.

[Editors' note: further revisions were suggested prior to acceptance, as described below.]

Thank you for resubmitting your work entitled "Developmental single-cell transcriptomics of hypothalamic POMC neurons reveal the genetic trajectories of multiple neuropeptidergic phenotypes" for further consideration by *eLife*. Your revised article has been reviewed by 2 peer reviewers and the evaluation has been overseen by Marianne Bronner as the Senior Editor, and a Reviewing Editor.

The revisions by the authors improved the manuscript. These data will be a valuable resource for the field. However, there are a few important points that should be corrected or clarified in a final version.

1. Characterization of Ebf1 expression

The claim that Ebf1 expression at E11.5 is localized to the mantle layer (line 201) and the labels in the accompanying Figure 1E are not accurate. Expression of Pomc and Ebf1 are located within the ventricular zone (VZ), albeit toward the lateral edge, as indicated by the dense DAPI stain. The mantle layer "ML" is label is not accurate; there is almost no "mantle layer" in the mediobasal hypothalamus at E11.5. Also, the differentiating zone indicated at the lateral edge of the VZ in Figure 1E is not striatal.

Note that Figure 2-Figure Suppl 1A shows co-expression of Pomc and Ebf1 at the border between the VZ and ML at E13.5. There appears to be more co-expression on the VZ side than the ML side. The expression pattern shown here is consistent with reports in other regions of the developing CNS, where EBFs have been implicated in regulating migration out of the VZ.

2. Lines 474-5 "The diffusion maps analysis appears to confirm the multiple origins of POMC progenitor cells demonstrated by the RNA velocity study."

This is a confusing sentence because "POMC progenitors" is not the accurate terminology to use. It is easier to just eliminate this sentence.

3. Discussion of Ebf1+/cluster 2 neurons

Lines 614-6 "The expression profile of cluster 2.Pomc^(med)^614 /Ebf1 differs from cluster 1.Pomc^(high)^615 /Prdm12 most particularly at the earliest developmental ages E11.5 to E15.5 and may both represent alternative genetic routes leading to mature POMC neurons."

There is no evidence that the Ebf1+/cluster 2 neurons develop into mature neurons of any sort. The analysis in Figure 6 depicts the Ebf1+/cluster 2 neurons as a dead end in Ray 1 (i.e. retains immature properties). Since the transcriptomics performed here relied on the expression of a Pomc-GFP transgene, it is possible that cluster 2 neurons differentiate into mature non-POMC cell types. The sentence should be corrected by swapping "arcuate" for "POMC".

4. Discussion of developmental origins of NPY and KNDy neurons Lines 619-626 are confusing.

Timing question: With regard to NPY and KNDy neurons, which are known to derive from Pomc-expressing neurons, the authors argue that they arise "earlier than expected "(presumably at E11.5). And yet, according to the developmental clustering (Figure 4) and developmental trajectory analyses (Figure 6 supplement 1), meaningful numbers of Otp+/cluster 4 (future NPY) and Tac2+/cluster 5 (future KNDy) neuron precursors emerge at E13.5, consistent with reports in the literature.

Developmental lineage: These analyses also support the idea that Otp+/cluster 4 (future NPY) neurons share the same developmental trajectory as Prdm12+/cluster 1 (future POMC) neurons (Figure 6) as well as a common precursor (E13.5-4, Figure 4). This should be explicitly discussed. These studies do not provide clues into the origin of the Tac2+/cluster 5 (future KNDy) neurons.

---

## [Author Response]

Essential revisions:1) Showing in situ hybridization data for E11.5 markers in a coronal plane, where the ventricular and mantle layer can de distinguished, would help to resolve the question of whether Ebf-hi expression represents newly differentiated neurons that are in the process of migrating out of the ventricular zone. (This appears to be case from the E13.5 section in Figure 1F, but it is hard to tell.)

As suggested by reviewer #2, we performed in situ hybridization studies using antisense RNAscope probes to simultaneously label *Pomc* and *Ebf1* expression in coronal sections of E11.5 embryos at the level of the developing basal hypothalamus. Using this additional embryonic time point and plane sectioning (in the original manuscript we had shown sagittal sections of E13.5 mouse embryos), we confirmed that *Ebf1* and *Pomc* co-express in many cells located in the mantle zone of the developing hypothalamus, as can be observed in the newly submitted Figure 2E and Figure 2—figure supplement 1. The previously shown Figure 2F is now included in Figure 2—figure supplement 1. As a consequence of including this new supplement figure, the previous Figure 2—figure supplement 1 is now named Figure 2—figure supplement 2 and its reference changed in the Results section (line 184-195). It is worth mentioning that the level of *Ebf1* expression in the developing hypothalamus is much lower than that found in other areas of the same sections, such as the striatal differentiating zone (SDZ) and mesenchyme (MES) as clearly observed in Figure 2E. Our scRNA-seq study shows that the expression level of *Ebf1* in POMC cells is highest at E11.5, diminishes at E13.5, is even further reduced at E15.5 and completely disappears from E17.5 and beyond (Figure 2D). These results are confirmed by in situ hybridization (Figure 2E and Figure 2—figure supplement 1). Because *Ebf1* has been involved in the control of cell differentiation and migration of early neurons from the subventricular zone to the mantle in the developing striatum (Garel, Marin, Grosschedl, and Charnay, 1999), it is tempting to speculate that *Ebf1* plays a similar role in early *Pomc*-expressing hypothalamic cells. However, our data precludes us from making such a conclusion at this point and further studies will be necessary to evaluate the functional role of *Ebf1* during the ontogeny of hypothalamic POMC neurons.

2) Addition of scRNA-seq data from earlier embryonic stages (i.e. E10.5 and/or E11.0) would help to identify the precursor to the population of immature POMC neurons that is captured by cluster 1.POMC-hi/Prdm12.

The reviewer points to a very interesting question which in our hands turned up to be technically more challenging than expected. At the initial design of this study, we sought to FACS cells from E10.5 developing hypothalami to capture the transcriptome at the onset of *Pomc* expression. However, the number of fluorescent cells in *Pomc*-dsRed transgenic E10.5 embryos was very small probably due to the existing delay between the onset of *Pomc*-driven transcription of the transgene and the translation of the early dsRed mRNAs into the fluorescent protein Tdimer2. In our hands, we need at least 17,000 cells to finally obtain 10,000 FACS cells for sequencing. In our current study, primary cells from POMC-TdDsRed/+ mice were first FACS sorted for the transgenic red fluorophore. The sorting percentage for POMC-TdDsRed cells is only 1-2%. Thus, it is very challenging to collect enough cells for FACS followed by scRNA-seq library preparation. We have previously reported using either an anti- ACTH antibody or anti-EGFP antibody in Pomc-EGFP E10.5 embryos that the number of cells expressing proteins from the *Pomc* gene or *Pomc* transgenes are very limited at this stage (Nasif et al., 2015, PNAS, Figure 1A and Hael et al., 2020 Mol Metab, Figure 1) (Hael, Rojo, Orquera, Low, and Rubinstein, 2020; Nasif et al., 2015).

3) One question raised by these data are the presence of an inhibitory population of GABA releasing (ie Vgat-expressing) POMC neurons. This is controversial with some studies reporting this and others indicating that there are no GABAergic POMC neurons. Given the evidence for a developmental cluster that is evident even at P12, it is important to resolve this issue in the context of the authors report of a bona fide transcriptional cluster with these co-expression patterns. The authors should examine the colocalization of these Pomc/Vgat neurons using RNA-FISH and report their distribution across development.

Typically, neurons are classified as either glutamatergic based on their expression of one or more vesicular glutamate transporters VGLUT1, VGLUT2 and VGLUT3 mRNAs (*Slc17a7, Slc17a6 and Slc17a8,* respectively) or GABAergic based on their expression of the vesicular GABA and glycine transporter VGAT (*Slc32a1*). Alternatively, GABAergic neurons are identified by expression of the synthetic enzymes glutamate decarboxylase 1 or 2 (*Gad67* and *Gad65*, respectively). None of these three last mentioned markers can definitively identify a GABAergic neuron and we do not report the coexpression of *Pomc* and *Slc32a1* in our manuscript. Moreover, triple in situ hybridization studies have demonstrated that subpopulations of POMC neurons express both *Slc17a6* and *Gad65* and the proportions of these neurons change during early postnatal life in the mouse (Jones et al., 2019; Wittmann, Hrabovszky, and Lechan, 2013). Synaptic storage and release of GABA from neuronal terminals after appropriate stimulation is in our view the definitive means to identify a neuron as GABAergic. Several publications have shown by E-physiology that some subpopulations of POMC neurons induce ipsps that can be blocked by GABA receptor antagonists (Hentges, Otero-Corchon, Pennock, King, and Low, 2009). Although VGAT expression has indeed not been found in POMC neurons, they do express VMAT as shown in Figure 3—figure supplement 1 and VMAT can act as an alternative synaptic GABA vesicular transporter in dopamine neurons (German, Baladi, McFadden, Hanson, and Fleckenstein, 2015).

4) Finally, several conclusions need to be tempered. Nonetheless, the paper will be an outstanding resource for the field. The discussion should be written to reflect the important resource of this data set.

As suggested by the reviewer we shortened some paragraphs of the Discussion to focus on the most important and less speculative matters. We have also tempered some conclusions and highlighted the importance and usefulness of this resource for future studies by many investigators worldwide. All changes in the revised Discussion are indicated in the revised manuscript.

Reviewer #1 (Recommendations for the authors):There seemed to be a substantial subset of microglia expressing Pomc (Figure 1), did the authors remove these from the dataset? How do they determine a bona fied Pomc neuron vs a neuron with Pomc mRNA contamination.

A general caveat of the scRNA sequencing technique is the potential contamination with RNA molecules from other cells present in the sample or even from the environment. Although this possibility can never be completely ruled out, in our study we used fluorescence assisted cell sorting from medial basal hypothalami collected from POMC-DsRed transgenic mice to enrich the population of cells to be analyzed at the single cell level. In addition, we took several levels of precaution to limit, monitor and discard potential cases of cross contamination events: (1) We applied stringent filtering criteria to exclude doublets and low quality cells as described in the Materials and methods section; (2) We performed an independent unsupervised analysis on cells expressing the transgenic red fluorophore (DsRed) while ignoring *Pomc* UMI criteria per cell (Figure1—figure supplement 2 and Figure 1—figure supplement 3) and we found similar clustering at each developmental age compared to clustering based on *Pomc* UMI criteria of > = 1. (3) To validate each cluster, we performed dual RNAscope in situ hybridization to confirm the colocalization of *Pomc* and feature genes from each cluster (Figure 2, Figure 3 and their linked supplemental figures). The glial cell clusters in this study were defined by showing simultaneous expression of high levels of several marker genes selectively representing microglia, VLMC, or Astrocytes (Figure 1C), and the absence of neuronal markers whereas a reciprocal criterion was used to define neuronal clusters. Because the current study focuses on *Pomc* expressing neurons, we have only studied and reported neuronal clusters (1-8) in greater detail.

A supplemental model figure/cartoon integrating the various developmental trajectories with their defining transcriptional factors would be useful to visualize the key findings of the paper.

The reviewer’s suggestion is a genuine interest we had for a long time. Unfortunately, we have been unable to capture in a single schematic the design and most meaningful results of this complex work without duplicating graphs already shown in Figures 1, 2, 3 and 6 of this manuscript.

Reviewer #2 (Recommendations for the authors):• Showing in situ hybridization data for E11.5 markers in a coronal plane, where the ventricular and mantle layer can de distinguished, would help to resolve the question of whether Ebf-hi expression represents newly differentiated neurons that are in the process of migrating out of the ventricular zone. (This appears to be case from the E13.5 section in Figure 1F, but it is hard to tell.)

This question was answered in the Essential Revision 1

• Addition of scRNA-seq data from earlier embryonic stages (i.e. E10.5 and/or E11.0) would help to identify the precursor to the population of immature POMC neurons that is captured by cluster 1.POMC-hi/Prdm12.

This question was answered in the Essential Revision 2.

• The number of total neurons changes dramatically across gestation, in large part because cells that adopt non-POMC fates turn off Pomc expression and are no longer captured by FACS. In Figure 1, reporting the absolute number of neurons in each cluster, instead expressing it as a percentage, would make it easier to interpret.

We appreciate the reviewer’s suggestion and have edited Figure 1 accordingly.

• Analyses shown in Figure 3 Supplemental Figures 3-4 are critical to resolve issues related to POMC progenitor heterogeneity. This information should be moved to the main figures. Maintaining a consistent color scheme between Figure 1 and Figure 3 Supplement 3 would make it much easier to interpret this data. The chart in Supplemental Figure 4 could then remain as a supplemental figure to reinforce this point.

We thank the reviewer’s suggestion and have edited Figure 3 Supplemental Figure3-4 accordingly in this revised version of the manuscript.

• Trajectory analyses shown in Figure 5 should include separate analyses of 2.Pomc-med/Ebf1 and 3.Pomc-med/Nra2, as they appear to map onto different populations in the mature ARH (Supplemental Figure 15).

We agree with the reviewer that the interpretation of original Supplementary Figure15 is not straightforward and may be misleading. Therefore, we decided to remove this comparative analysis from the Results section and its corresponding figures. Based on the reviewer’s comment, the original Figure 5 became Figure 6.

• There are discrepancies between the text (lines 489-497) and Supplementary file 14, which do not link 3.Pomc-med/Nr4a2 neurons to mature POMC populations and Supplementary file 15, which does, making it difficult to interpret these data.

This part has been removed from the Results section.

References

Garel, S., Marin, F., Grosschedl, R., & Charnay, P. (1999). Ebf1 controls early cell differentiation in the embryonic striatum. *Development, 126*(23), 5285-5294.

German, C. L., Baladi, M. G., McFadden, L. M., Hanson, G. R., & Fleckenstein, A. E. (2015). Regulation of the Dopamine and Vesicular Monoamine Transporters: Pharmacological Targets and Implications for Disease. *Pharmacol Rev, 67*(4), 1005-1024. doi:10.1124/pr.114.010397

Hael, C. E., Rojo, D., Orquera, D. P., Low, M. J., & Rubinstein, M. (2020). The transcriptional regulator PRDM12 is critical for Pomc expression in the mouse hypothalamus and controlling food intake, adiposity, and body weight. *Mol Metab, 34*, 43-53. doi:10.1016/j.molmet.2020.01.007

Hentges, S. T., Otero-Corchon, V., Pennock, R. L., King, C. M., & Low, M. J. (2009). Proopiomelanocortin expression in both GABA and glutamate neurons. *J Neurosci, 29*(43), 13684-13690. doi:10.1523/JNEUROSCI.3770-09.2009

Jones, G. L., Wittmann, G., Yokosawa, E. B., Yu, H., Mercer, A. J., Lechan, R. M., & Low, M. J. (2019). Selective Restoration of Pomc Expression in Glutamatergic POMC Neurons: Evidence for a Dynamic Hypothalamic Neurotransmitter Network. *eNeuro, 6*(2). doi:10.1523/ENEURO.0400-18.2019

Nasif, S., de Souza, F. S., Gonzalez, L. E., Yamashita, M., Orquera, D. P., Low, M. J., & Rubinstein, M. (2015). Islet 1 specifies the identity of hypothalamic melanocortin neurons and is critical for normal food intake and adiposity in adulthood. *Proc Natl Acad Sci U S A, 112*(15), E1861-1870. doi:10.1073/pnas.1500672112

Wittmann, G., Hrabovszky, E., & Lechan, R. M. (2013). Distinct glutamatergic and GABAergic subsets of hypothalamic pro-opiomelanocortin neurons revealed by in situ hybridization in male rats and mice. *J Comp Neurol, 521*(14), 3287-3302. doi:10.1002/cne.23350

[Editors' note: further revisions were suggested prior to acceptance, as described below.]

The revisions by the authors improved the manuscript. These data will be a valuable resource for the field. However, there are a few important points that should be corrected or clarified in a final version.1. Characterization of Ebf1 expressionThe claim that Ebf1 expression at E11.5 is localized to the mantle layer (line 201) and the labels in the accompanying Figure 1E are not accurate. Expression of Pomc and Ebf1 are located within the ventricular zone (VZ), albeit toward the lateral edge, as indicated by the dense DAPI stain. The mantle layer "ML" is label is not accurate; there is almost no "mantle layer" in the mediobasal hypothalamus at E11.5. Also, the differentiating zone indicated at the lateral edge of the VZ in Figure 1E is not striatal.Note that Figure 2-Figure Suppl 1A shows co-expression of Pomc and Ebf1 at the border between the VZ and ML at E13.5. There appears to be more co-expression on the VZ side than the ML side. The expression pattern shown here is consistent with reports in other regions of the developing CNS, where EBFs have been implicated in regulating migration out of the VZ.

In agreement with the reviewer’s comment, we have modified the sentence (line 186-189) indicating now that the co-expression of *Ebf1* and *Pomc* occurs at the ventricular zone. Accordingly, we have modified the labels in Figure 2E. Concerning the reviewer’s suggestion that *Ebf1* may regulate cell migration out of the VZ during neuronal development, we have now include this hypothesis in the Discussion section (lines 514 – 517).

2. Lines 474-5 "The diffusion maps analysis appears to confirm the multiple origins of POMC progenitor cells demonstrated by the RNA velocity study."This is a confusing sentence because "POMC progenitors" is not the accurate terminology to use. It is easier to just eliminate this sentence.

Following the reviewer’s suggestion we have deleted this sentence in the newly revised manuscript.

3. Discussion of Ebf1+/cluster 2 neuronsLines 614-6 "The expression profile of cluster 2.Pomc^(med)^614 /Ebf1 differs from cluster 1.Pomc^(high)^615 /Prdm12 most particularly at the earliest developmental ages E11.5 to E15.5 and may both represent alternative genetic routes leading to mature POMC neurons."There is no evidence that the Ebf1+/cluster 2 neurons develop into mature neurons of any sort. The analysis in Figure 6 depicts the Ebf1+/cluster 2 neurons as a dead end in Ray 1 (i.e. retains immature properties). Since the transcriptomics performed here relied on the expression of a Pomc-GFP transgene, it is possible that cluster 2 neurons differentiate into mature non-POMC cell types. The sentence should be corrected by swapping "arcuate" for "POMC".

Following the reviewer’s suggestion, we have replaced “POMC” with “arcuate” in the newly revised manuscript.

4. Discussion of developmental origins of NPY and KNDy neurons Lines 619-626 are confusing.Timing question: With regard to NPY and KNDy neurons, which are known to derive from Pomc-expressing neurons, the authors argue that they arise "earlier than expected "(presumably at E11.5). And yet, according to the developmental clustering (Figure 4) and developmental trajectory analyses (Figure 6 supplement 1), meaningful numbers of Otp+/cluster 4 (future NPY) and Tac2+/cluster 5 (future KNDy) neuron precursors emerge at E13.5, consistent with reports in the literature.

We thank the reviewer’s constructive suggestion and have revised this sentence accordingly (lines 520 – 523).

We would like to remark that POMC cells from cluster 4 display relatively high levels of *Ot*p already at E11.5. Although *Npy* and *Agrp* transcripts only peak up days later, the early presence of *Otp* at E11.5 is likely an active marker for the differentiation pathway leading to the AGRP phenotype. We would also like to point out that developmental subclustering of POMC cells only at E11.5 revealed a small but distinct group of related cells (E11.5-9) featuring *Otp* (Figure 4-Source Data2). This subcluster shares high similarity to *Pomc*^(low)^*/Otp* cluster in regards to the gene expression profiles (comparing Figure 4-Source Data 2 to Figure 1-Source Data 2). In situ hybridization studies shown by the Allen Developmental Mouse Brain Atlas also confirmed the expression of *Otp* in the developing hypothalamus at E11.5 and it is likely these early *Pomc*^(low)^*/Otp* cells are the precursors of AGRP/NPY neurons at later embryonic stages, although this hypothesis requires further investigation.

Developmental lineage: These analyses also support the idea that Otp+/cluster 4 (future NPY) neurons share the same developmental trajectory as Prdm12+/cluster 1 (future POMC) neurons (Figure 6) as well as a common precursor (E13.5-4, Figure 4). This should be explicitly discussed. These studies do not provide clues into the origin of the Tac2+/cluster 5 (future KNDy) neurons.

We thoroughly examined our raw data. We agree that the *Pomc*^(low)^*/Otp* neurons share the common precursors as *Pomc*
^(high)^ / *Prdm12* cluster, as shown by the RNA-velocity trajectory analysis in Figure 6A and Figure 6—figure supplement 1B and by the transcriptomic analysis at individual developmental stages (Figure 4—figure supplement 1). We have now addressed this issue in the revised manuscript (Line 528-530).

The reviewer pointed out the existence of common precursor (E13.5-4, Figure 4), indicating the *Otp+*/cluster 4 (future NPY) neurons share the same developmental trajectory as *Prdm12+*/cluster 1 (future POMC) neurons. We have carefully re-examined the Figure 4 and its associated supplement figures and source data and confirmed that the gene profiles in E11.5-9 and E13.5-4 subclusters are highly similar to that of *Pomc*^(low)^*/Otp* cluster with the feature genes – *Otp*, *Calcr*, and *Sst*. We also found these two clusters share genes that are highly expressed in *Pomc*
^(high)^ / *Prdm12,* such as *Dlk1, Six3, Six6, Cited1, Isl1 and Peg3*. That’s why we consider E11.5-9 and E13.5-4 subclusters as the common origins for both *Pomc*^(low)^*/Otp* and *Pomc*
^(high)^ / *Prdm12* clusters (Figure 4—figure supplement 1). It worth noting that AGRP neurons share several similarities with POMC neurons in terms of transcription profiles although their physiological functions are opposite. For example, *Tbx3* is expressed in both AGRP and POMC neurons, although its role appears to be different in each cell type based on the phenotype observed in the individual conditional mouse mutants.